# Approximations to the Fisher Information Metric of Deep Generative Models for Out-Of-Distribution Detection

**Sam Dauncey***                                                                                   *s.dauncey@sms.ed.ac.uk*
*Department of Mathematics*
*University of Edinburgh*

**Chris Holmes**                                                                                   *cholmes@stats.ox.ac.uk*
*Department of Statistics*
*University of Oxford*
*The Alan Turing Institute*

**Christopher Williams**                                                                     *williams@stats.ox.ac.uk*
*Department of Statistics*
*University of Oxford*

**Fabian Falck**                                                                               *fabian.falck@stats.ox.ac.uk*
*Department of Statistics*
*University of Oxford*
*The Alan Turing Institute*

*\* Work partially done during an internship at University of Oxford.*

**Reviewed on OpenReview:** *https: // openreview. net/ forum? id=EcuwtinFs9*

## Abstract

Likelihood-based deep generative models such as score-based diffusion models and variational autoencoders are state-of-the-art machine learning models approximating high-dimensional distributions of data such as images, text, or audio. One of many downstream tasks they can be naturally applied to is out-of-distribution (OOD) detection. However, seminal work by Nalisnick et al. which we reproduce showed that deep generative models consistently infer higher log-likelihoods for OOD data than data they were trained on, marking an open problem. In this work, we analyse using the gradient of a data point with respect to the parameters of the deep generative model for OOD detection, based on the simple intuition that OOD data should have larger gradient norms than training data. We formalise measuring the size of the gradient as approximating the Fisher information metric. We show that the Fisher information matrix (FIM) has large absolute diagonal values, motivating the use of chi-square distributed, layer-wise gradient norms as features. We combine these features to make a simple, model-agnostic and hyperparameter-free method for OOD detection which estimates the joint density of the layer-wise gradient norms for a given data point. We find that these layer-wise gradient norms are weakly correlated, rendering their combined usage informative, and prove that the layer-wise gradient norms satisfy the principle of (data representation) invariance. Our empirical results indicate that this method outperforms the Typicality test for most deep generative models and image dataset pairings.

## 1 Introduction

Neural networks can be highly confident but incorrect when given inputs different to the distribution of data they were trained on (Szegedy et al., 2014; Nguyen et al., 2015). While domain generalisation Zhou et al. (2021) and domain adaptation (Garg et al., 2023; Ganin et al., 2016) methods tackle this problem by learning

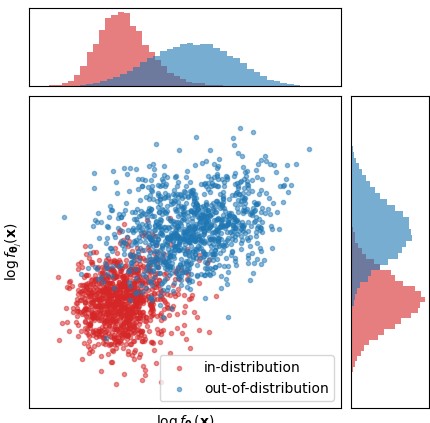

Figure 1: *Gradients from certain layers are highly informative for OOD detection.* We select two highly informative neural network layers of a deep generative model with parameters $\boldsymbol{\theta}_i, \boldsymbol{\theta}_j$ from a Glow Kingma & Dhariwal (2018) model trained on `CIFAR-10` and plot the $L^2$-norm of the gradients $f_{\boldsymbol{\theta}_j} = \left\| \nabla_{\boldsymbol{\theta}_j} l(\boldsymbol{x}) \right\|$ for in-distribution (`CIFAR-10`) and out-of-distribution (`SVHN`) samples $\boldsymbol{x}$. The gradient norm of these two layers allows to separate in- and out-of-distribution samples.

machine learning systems which are robust or can actively adapt to domain shift, there may be scenarios where for specific data points this domain shift is too severe to draw reliable inferences. Identifying and possibly filtering out anomalies or *out-of-distribution (OOD)* inputs before deploying the model in the wild is a viable strategy in such cases, especially for safety-critical applications (Ulmer et al., 2020; Stilgoe, 2020; Baur et al., 2021).

*Deep generative models* such as variational autoencoders Kingma & Welling (2014), normalising flows Papamakarios et al. (2021), autoregressive models Van den Oord et al. (2016); Salimans et al. (2017) and diffusion models Sohl-Dickstein et al. (2015b); Ho et al. (2020) are an important family of models in machine learning which allow us to generate high-quality samples from high-dimensional, multi-modal conditional or unconditional distributions in domains such as images, videos, text and speech. Many current state-of-the-art methods are probabilistic: they approximate the data log-likelihood, the likelihood of a data sample given the learned model parameters under the data distribution, or a lower-bound thereof. This renders them a natural candidate for the task of OOD detection as they 'out-of-the-box' provide an OOD metric Bishop (1994) (i.e. the approximated data likelihood) which they use the training objective. However, Nalisnick et al. (2019a); Choi et al. (2018) showed that many of the above mentioned classes of probabilistic deep generative models consistently infer *higher* log-likelihoods for data points drawn from OOD datasets (non-training data) than for in-distribution (training data) samples. In Fig. 2 we replicated their results with a Glow model, and show that score-based diffusion models are likewise affected by the phenomenon. This result is very surprising given that generative models are trained to maximise the log-likelihood of the training data, and are able to generate high-fidelity, diverse samples from the training distribution. This marks an open problem for deep generative models, and renders the direct use of their estimated likelihood in out-of-distribution detection infeasible. It also questions what deep generative models learn during training, and how they generalise.

Related work tackled this problem from two angles: explaining why log-likelihood estimates of these models fail to discriminate, Kirichenko et al. (2020); Zhang et al. (2021a); Le Lan & Dinh (2021); Caterini & Loaiza-Ganem (2022), and proposing likelihood-based OOD detection methods and adaptations of existing ones which may overcome these shortcomings Ren et al. (2019); Choi et al. (2018); Hendrycks et al. (2018); Liu et al. (2020); Havtorn et al. (2021); Nalisnick et al. (2019b). In §4 we will analyse the previous work on gradient-based OOD detection, linking independent discoveries of other authors Nguyen et al. (2019); Kwon et al. (2020); Choi et al. (2021); Bergamin et al. (2022).

**Motivation and Intuition.** This paper presents an alternative approach for OOD detection which we motivate in the following. Consider the example of a (linear) regression model fitted on some (in-distribution) training data. If we now include an outlier in our training data and refit the model, the outlier will have a lot of influence on our estimate of the model's parameters compared to other in-distribution data points. One way to formalise this intuition is using the *hat-value*: The *hat-value* is defined as the derivative $\frac{d\hat{y}}{dy}$ of the

model's prediction $\hat{y}$ with respect to a given (dependent) data point $y$. It describes the leverage of a single data point, which is used for fitting the model, on the model's prediction for that data point after fitting.

We extend this intuition of OOD to deep learning by considering the gradient of the log-likelihood of a deep generative model with respect to its parameters, also known as the *score*. If a neural network converged to a local (or global) minimum, we expect the gradient of the likelihood with respect to the model's parameters to be flat for training data points. Hence, the score is small in norm, and performing an optimiser step for a training data point would not change its parameters much. If after many epochs of training we were now to present the model an OOD data point which it has not seen before, we expect the gradient—resembling the 'hat-value of a neural network'—to be steep: the norm of the score would be large, just like hat values are large (in absolute value) for OOD data. An optimizer step with an OOD data point would change the neural network's parameters a lot. It is this intuition which motivates us to theoretically analyse the use of the gradient for OOD detection. In preview of our analyses, in Fig. 1 (and later more thoroughly in Fig. 4) we will precisely observe that (layer-wise) gradient norms are in general larger for OOD than for in-distribution data, which enables the use of gradients for OOD detection. Code to reproduce our experimental results is publicly available on GitHub [1].

Our contributions are as follows: (a) We analyse the use of the gradient of a data point with respect to the parameters of a deep generative model for OOD detection, and formalise this as approximating the Fisher information metric, a natural way of measuring the size of the gradient. (b) We show that the Fisher information matrix (FIM) has large absolute diagonal values, motivating the use of layer-wise gradient norms which are chi-square distributed as a possible approximation. Our theoretical results show that layer-wise gradients satisfy the principle of (data representation) invariance Le Lan & Dinh (2021), a desirable property for OOD methods. We also find that these layer-wise gradient norms are weakly correlated, making their combined usage more informative. (c) We propose a first simple, model-agnostic and hyperparameter-free method which estimates the joint density of layer-wise gradient norms for a given data point. In our experiments, we find that this method outperforms the Typicality test for most deep generative models and image dataset pairings.

## 2 Current Methods for OOD detection

In this section, we define the OOD detection problem, describe the open problem of using deep generative models for OOD detection and how the input representation may explain this, and how a gradient-based method can be a compelling approach which is invariant to the input representation.

### 2.1 OOD detection: Problem Formulation

Given training data $\boldsymbol{x}_1 \dots \boldsymbol{x}_N$ drawn from a distribution $p$ over the input space $\mathcal{X} \subseteq \mathbb{R}^D$, we define the problem of OOD detection as assigning an OOD score $S(\boldsymbol{x})$ to each $\boldsymbol{x} \in \mathcal{X}$ such that points with low OOD scores are semantically similar to points sampled from $p$. OOD detection is *unsupervised* if it is not given class label information at training time.

The specific problem we are interested in is leveraging recent advances in deep generative models for unsupervised OOD detection. Here a deep generative model $p^{\boldsymbol{\theta}}$ is trained to approximate the distribution of some training data $\boldsymbol{x}_1 \dots \boldsymbol{x}_N \sim p$, and $S$ is a statistic derived from $p^{\boldsymbol{\theta}}$ (such as the model likelihood Nalisnick et al. (2019b), a latent variable hierarchy Schirrmeister et al. (2020); Havtorn et al. (2021), or combinations thereof Morningstar et al. (2021)).

In order to evaluate an OOD detection method, one is required to select semantically dissimilar surrogate out-distributions (e.g. a different dataset) to test against. Previous work has sought to define OOD detection as a generic test against data sampled from any differing distribution Hendrycks & Gimpel (2017). Our additional requirement that the out-distribution is semantically dissimilar is motivated by recent theoretical work by Zhang et al. (2021a) showing that a single-sample test against all out-distributions is impossible.

---

[1] https://github.com/SamD770/Generative-Models-Knowledge

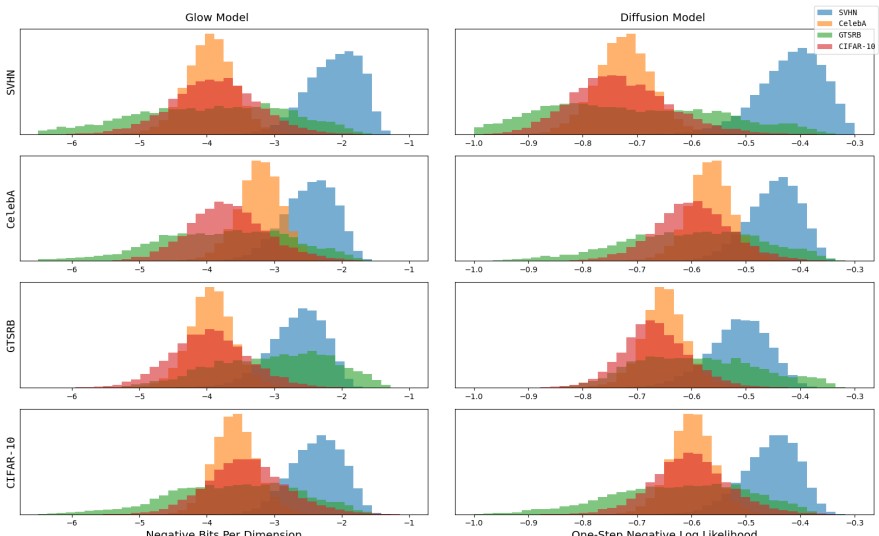

Figure 2: *Counter-intuitive properties of likelihood-based generative models.* Histogram of the negative log-likelihoods inferred from a Diffusion Ho et al. (2020) model [Left] and a Glow Kingma & Dhariwal (2018) model [right] trained on one of four image datasets (corresponding to the four subplots) and evaluated on the test set of all four datasets, respectively. For diffusion models we use the negative log-likelihood from one step of the diffusion process $p^{\boldsymbol{\theta}}(\boldsymbol{x_0}|\boldsymbol{x_1})$. For both models we scale the log-likelihoods by the dimensionality of the data, in this case $3 \times 32 \times 32$. This Figure replicates the results in the seminal paper by Nalisnick et al. (2019a), noting that our results for diffusion models are novel. We find that the training dataset has a counter-intuitively small impact on the ordering of the datasets as ranked by log-likelihood.

## 2.2 Likelihood-based methodology for unsupervised OOD detection

**Likelihood thresholding.** Bishop (1994) proposed using the learned model's negative log likelihood as an OOD score $S(\boldsymbol{x}) = -\log p^{\boldsymbol{\theta}}(\boldsymbol{x})$. In their seminal paper, Nalisnick et al. empirically demonstrated that this approach fails for a wide variety of deep generative models (Nalisnick et al., 2019a). In particular they showed that certain image datasets such as SVHN are assigned systemically higher likelihoods than other image datasets such as CIFAR10, independent of the training distribution. We replicate this result (for a Glow model, a type of normalising flow, and for the first time a denoising diffusion model) in Figure 2. In their follow up work Nalisnick et al. argue that, in the example of a standard Gaussian of large dimension $D$, samples close to the origin should be classified as OOD as the Gaussian annulus result Blum et al. (2020) demonstrates that the vast majority of samples from have a distance of $\sqrt{D}$ from the origin. Generalising this to generative models, they argue that samples with likelihoods much higher than likelihoods of in-distribution samples must be semantically atypical (Nalisnick et al., 2019b). They use this to motivate OOD scoring based on the likelihood being too high or too low, defining the typicality Cover & Thomas (1991) score as $S(\boldsymbol{x}) = |\log p^{\boldsymbol{\theta}}(\boldsymbol{x}) - \hat{\mathbb{H}}|$ , where $\hat{\mathbb{H}}$ is the average log-likelihood on some held-out training data.

**Likelihood ratios.** The likelihood assigned by deep generative models has been shown to strongly correlate with complexity metrics such as the compression ratio achieved by simple image compression algorithms Serrà et al. (2020), and likelihoods from other generative models trained on highly diverse image distributions Schirrmeister et al. (2020), with the highest likelihoods being assigned to constant images. This is somewhat expected, as for discrete data the negative log likelihood is directly proportional to the number of bits needed to encode the data under arithmetic coding. To add to these findings, in Appendix A.3 we use a very simple complexity metric $TV$, the total variation achieved by considering the image as a vector in $[0, 1]^{784}$, to show that the whole of MNIST is contained in a set of bounded complexity with volume (Lebesgue measure) $10^{-116}$. Thus a model needs to only assign a very low prior probability mass to this set for high likelihoods to be

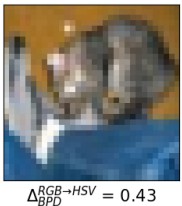 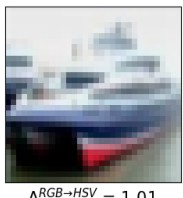

$\Delta_{BPD}^{RGB \to HSV} = 0.43$     $\Delta_{BPD}^{RGB \to HSV} = 1.01$

Figure 3: *The log-likelihood heavily depends on data representation* Le Lan & Dinh (2021). Here we plot the first two samples of the `CIFAR10` dataset and the difference in Bits Per Dimension (BPD) induced by changing from an RGB to an HSV colour model:

$$\Delta_{BPD}^{RGB \to HSV} = \frac{\log_2 p_{RGB}(\mathbf{x}) - \log_2 p_{HSV}(\mathbf{x})}{3 \times 32 \times 32}.$$

In Appendix B.1, we provide experimental details and inFig. 8 replicate this for the first 20 samples, where we observe $\Delta_{BPD}^{RGB \to HSV}$ values ranging from 0.18 to 1.76

achieved, demonstrating the important connection between volume, complexity and model likelihoods which we hence discuss in §2.3.

Ren et al. (2019) argue in favour of using likelihood ratio tests in order to factor out the influence of the "background likelihood", the model's bias towards assigning high likelihoods to images with low complexity. In practice, this requires modelling the background likelihood via corruption of training data Ren et al. (2019), out-of-the-box and neural compressors Serrà et al. (2020); Zhang et al. (2021b) or the levels of a model's latent variable heirarchy Schirrmeister et al. (2020); Havtorn et al. (2021), leading to restrictions for the data modalities or models to which the method can be applied to. In general there is a limited number of use cases whereby one can pre-specify the OOD distribution $p^{OOD}$ they are interested in well enough that they can evaluate the likelihood of the data under this OOD distribution $\log p^{OOD}(\mathbf{x})$ (which is necessary to evaluate the likelihood), without access to samples from this distribution at which point classification becomes a better-studied option.

### 2.3 Representation dependence of the likelihood

Le Lan & Dinh (2021) emphasise that the definition of likelihood requires choosing a method of assigning volumes to the input space $\mathcal{X}$. Specifically, datapoints could be represented as belonging to some other input space $\mathcal{T}$, linked via a smooth invertible coordinate transformation $T : \mathcal{X} \to \mathcal{T}$. The model probability density for a given datapoint $\boldsymbol{x} \in \mathcal{X}$, which we denote $p_{\mathcal{X}}^{\boldsymbol{\theta}}(\boldsymbol{x})$, will thus differ from the probability density $p_{\mathcal{T}}^{\boldsymbol{\theta}}(\boldsymbol{t})$ of the corresponding point $\boldsymbol{t} = T(\boldsymbol{x})$ by a factor of the Jacobian determinant of $T$ (Le Lan & Dinh, 2021):

$$p_{\mathcal{T}}^{\boldsymbol{\theta}}(\boldsymbol{t}) = p_{\mathcal{X}}^{\boldsymbol{\theta}}(\boldsymbol{x}) \left| \frac{\partial T}{\partial \boldsymbol{x}} \right|^{-1}. \tag{1}$$

The *volume element* $\left| \frac{\partial T}{\partial \boldsymbol{x}} \right|^{-1}$ describes the change of volumes local to $\boldsymbol{x}$ as it is passed through $T$. This term can grow or shrink exponentially with the dimensionality of the problem, making its effect counter-intuitively large. As an empirical example in the case of image distributions, in Fig. 3 and Appendix B.1 Fig 8 we consider the case of a change of color model $T^{RGB \to HSV}$ from a Red-Green-Blue (RGB) to Hue-Saturation-Value (HSV) representation. We compute the induced change in bits per dimension as a scaled log-value of the volume element $\Delta_{BPD}^{RGB \to HSV} = \frac{1}{3 \times 32 \times 32} \log \left| \frac{\partial T^{RGB \to HSV}}{\partial \boldsymbol{x}} \right|$ and report values for 20 non-cherry picked `CIFAR`-10 images ranging from 0.18 to 1.76. For comparison the average BPD reported in the seminal paper by Nalisnick et al. (2019a) was 3.46 on `CIFAR10`, compared to 2.39 on `SVHN` when evaluating with the same model. Hence, if we use the likelihood for OOD detection, whether we classify a sample as OOD or not may flip for some samples merely by changing how the data is represented.

Motivated by the strong impact of the volume element, Le Lan & Dinh (2021) propose a *principle of (representation) invariance*: given a perfect model $p^{\boldsymbol{\theta}^*}$ of the data distribution, the outcome of an unsupervised OOD detection method should not depend on how we represent the input space $\mathcal{X}$. In theory likelihood ratios are representation-invariant (Le Lan & Dinh, 2021), however in practice the method used to generate the background distribution often re-introduces dependence on the representation. For example Ren et al.

(2019) propose to generate the background distribution by re-sampling randomly chosen pixels as independent uniform distributions, re-introducing the notion of volume.

## 2.4 Invariance of the gradient under invertible transformations

To achieve a representation invariant OOD score (Le Lan & Dinh, 2021), we are thus motivated to quotient out the effect of the volume element in Eq. (8). We now present our first theoretical contribution, which shows that methods based on the gradient of the log-likelihood do precisely this.

**Proposition 1.** Let $p_{\mathcal{X}}^{\boldsymbol{\theta}}(\boldsymbol{x})$ and $p_{\mathcal{T}}^{\boldsymbol{\theta}}(\boldsymbol{t})$ be two probability density functions corresponding to the same model distribution $p^{\boldsymbol{\theta}}$ being represented on two different measure spaces $\mathcal{X}$ and $\mathcal{T}$. Suppose these representations encode the same information, i.e. there exists a smooth, invertible reparameterization $T : \mathcal{X} \to \mathcal{T}$ such that for $\boldsymbol{x} \in \mathcal{X}$ and $\boldsymbol{t} \in \mathcal{T}$ representing the same point we have $T(\boldsymbol{x}) = \boldsymbol{t}$. Then, the gradient vector $\nabla_{\boldsymbol{\theta}}(\log p^{\boldsymbol{\theta}})$ is invariant to the choice of representation, and in particular, $\nabla_{\boldsymbol{\theta}}(\log p_{\mathcal{T}}^{\boldsymbol{\theta}})(\boldsymbol{t}) = \nabla_{\boldsymbol{\theta}}(\log p_{\mathcal{X}}^{\boldsymbol{\theta}})(\boldsymbol{x})$.

**Proof.** See Appendix A.1. We prove analagous results for variational lower bounds (e.g. the ELBO of a VAE) in Appendix A.2.

**Remark 1.** Training a generative model $p^{\boldsymbol{\theta}_0}$ with initialisation parameters $\boldsymbol{\theta}_0$ with log-likelihood as the loss via gradient-descent produces training trajectories $\boldsymbol{\theta}_0, \boldsymbol{\theta}_1, \dots \boldsymbol{\theta}_N$ which are representation-invariant.

The interpretation of the above results is subtle. We would like to caution the reader by noting it does *not* mean that the inductive biases are discarded when the gradient is computed as inductive biases pertaining to distances between data points are frequently encoded in the parameter space. Further, remark 1 may explain why the likelihood can still be used to train deep generative models and allow them to generate convincing samples when using a gradient-based optimisation algorithm, even though the likelihood value itself appears uninformative for detecting if data is in-distribution.

## 3 Methodology

In this section, we develop a mathematically-principled method for gradient-based OOD detection.

### 3.1 Layer-wise gradients are highly informative and differ in size by orders of magnitudes

We are now interested in formulating a method which uses the intuitively plausible (see §1) and data representation-invariant (see §2.4) score $\nabla_{\boldsymbol{\theta}} l(\boldsymbol{x}) = \nabla_{\boldsymbol{\theta}}\{\log p^{\boldsymbol{\theta}}\}(\boldsymbol{x})$ for OOD detection. A naïve approach would be to measure the size of the score vector by computing the $L^2$ norm $\|\nabla_{\boldsymbol{\theta}} l(\boldsymbol{x})\|_2^2$ of the gradient Nguyen et al. (2019). We can view this $L^2$ norm as the directional derivative of the log-likelihood $\log p^{\boldsymbol{\theta}}$ in the direction of its own gradient $\nabla_{\boldsymbol{\theta}} l(\boldsymbol{x})$, which can be intuited as a measure of how much the model can learn about the given datapoint with one small gradient update.

In the following, we analyse this idea, demonstrating its limitations: We empirically find that the size of the norm of the score vector is dominated by specific neural network layers which the overall gradient norm cannot capture. In Fig. 4, we train deep generative models (here: Glow Kingma & Dhariwal (2018) and diffusion models Ho et al. (2020)) on a training dataset (here: `CelebA`). We then draw a batch of items from different evaluation datasets and compute the squared *layer-wise* $L^2$-norm of the gradients of the log-likelihood of a deep generative model with respect to the parameters $\boldsymbol{\theta}_j$ of the corresponding layer, i.e. $f_{\boldsymbol{\theta}_j}(\boldsymbol{x}_{1:B}) = \left\|\nabla_{\boldsymbol{\theta}_j}(\sum_{b=1}^{B} l(\boldsymbol{x}_b))\right\|_2^2$. The histrogrammes in the left two columns plot $f_{\boldsymbol{\theta}_j}(\boldsymbol{x}_{1:B})$ for each layer separately, the plots in the rightmost column shows their interaction in a scatterplot.

Two points are worth noting: We observe that for a given neural network layer (and different batches), the gradients are of a similar size, but *across* layers, the scale of the layer-wise gradient norms differs by orders of magnitudes. In particular, taking the norm over the entire score vector would overshadow the signal of layers with a smaller norm by those on much larger magnitudes. Second, note that the layer-wise gradient norms do *not* strongly correlate for randomly selected layers. In particular, one may find two layers with corresponding features $f_{\boldsymbol{\theta}_j}(\boldsymbol{x}_{1:B})$ and $f_{\boldsymbol{\theta}_k}(\boldsymbol{x}_{1:B})$ which allow us to separate training (in-distribution)

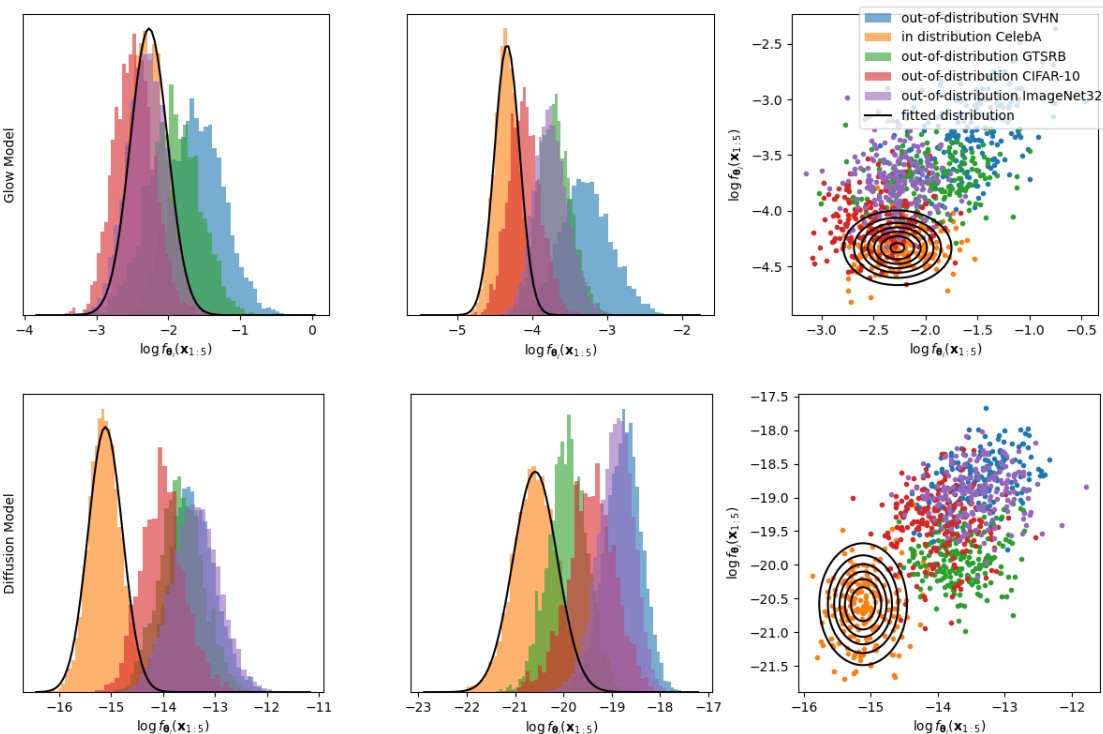

Figure 4: *Layer-wise gradients of the log-likelihood (the score) are highly informative for OOD detection.* Their size differs by orders of magnitudes between layers, and they are not strictly correlated, rendering layer-wise gradients (in contrast to the full gradient) discriminatory features for OOD detection. In each row, we randomly select two layers with parameters $\boldsymbol{\theta}_i, \boldsymbol{\theta}_j$ from a Glow Kingma & Dhariwal (2018) model [Top] or a Diffusion model Ho et al. (2020) [Bottom], which have 1353 and 276 layers, respectively. The models are trained on `CelebA`, a dataset that has proved challenging for OOD detection in previous work Nalisnick et al. (2019b). We then evaluate this model on batches $\boldsymbol{x}_{1:B}$ ($B = 5$) drawn from the in-distribution and OOD test datasets and compute the squared layer-wise $L^2$-norm of the gradients of the log-likelihood with respect to the parameters of the layer, i.e. $f_{\boldsymbol{\theta}_j}(\boldsymbol{x}_{1:B}) = \left\| \nabla_{\boldsymbol{\theta}_j} (\sum_{b=1}^{B} l(\boldsymbol{x}_b)) \right\|_2^2$. [Left and Middle] shows the two layer-wise gradients separately, [Right] shows their interaction in a scatter plot. In Appendix B Figures 9 - 11, we provide our complete results, showing more layers from three likelihood-based generative models, each trained and evaluated on five datasets.

from evaluation (OOD) data points with a line in this latent space. Considering an example, when fixing $f_{\boldsymbol{\theta}_1}(\boldsymbol{x}_{1:B}) \approx -7.5$, large negative values of $f_{\boldsymbol{\theta}_2}(\boldsymbol{x}_{1:B})$ are in-distribution, and as they become more positive, they correspond to the out-of-distribution datasets `CIFAR-10` and `ImageNet32`, respectively, with very high probability. This renders the layer-wise information superior over the overall gradient for use as discriminative features in OOD detection. We present our complete results in Appendix B Figs. 9-11, showing further layers (histogrammes), also for other deep generative models (VAEs Kingma & Welling (2014)) and training datasets (`SVHN`, `CelebA`, `GTSRB`, `CIFAR-10` & `ImageNet32`).

## 3.2 The Fisher Information Metric: A principled way of measuring the size of the gradient

Having identified the limitations of using the $L^2$ norm of the gradient, a perhaps mathematically more natural way to measure the score vector's size is to use the norm induced by the *Fisher information metric* $\|\nabla_{\boldsymbol{\theta}} l(\boldsymbol{x})\|_{FIM}$ (Radhakrishna Rao, 1948), defined as

$$\|\nabla_{\boldsymbol{\theta}} l(\boldsymbol{x})\|_{FIM}^2 = \nabla_{\boldsymbol{\theta}} l(\boldsymbol{x})^T F_{\boldsymbol{\theta}}^{-1} \nabla_{\boldsymbol{\theta}} l(\boldsymbol{x}), \qquad F_{\boldsymbol{\theta}} = E_{\boldsymbol{y} \sim p^{\boldsymbol{\theta}}} (\nabla_{\boldsymbol{\theta}} l(\boldsymbol{y}) \nabla_{\boldsymbol{\theta}} l(\boldsymbol{y})^T) \qquad (2)$$

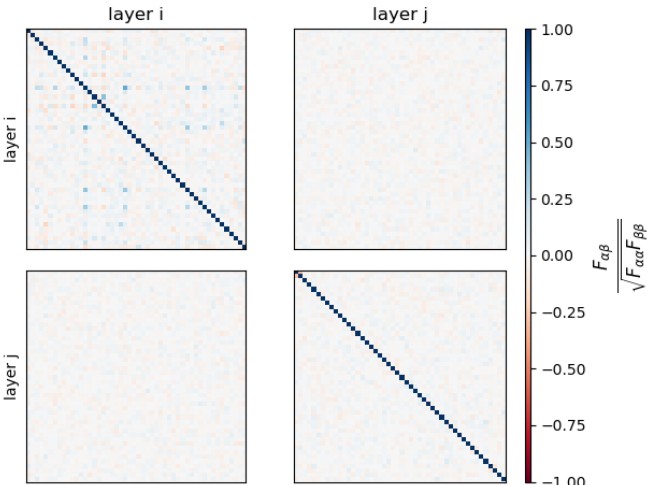

Figure 5: *The layer-wise FIM has large absolute diagonal values.* We randomly select two layers $\theta_i$ and $\theta_j$ from a Glow model trained on `CelebA`, and randomly select $\max(50, |\theta_j|)$ weights from each layer. We then compute slices of the FIM using the method described in Equation (3) and plot the results, with dark blue colours at coordinates $(\alpha, \beta)$ corresponding to larger values for the corresponding element of the FIM. In order to maintain visual fidelity of the plot when weights between layers vary by orders of magnitudes, we normalise row $\alpha$ by a factor of $\sqrt{F_{\alpha\alpha}}$ where $F_{\alpha\alpha}$ indicates the element of the FIM at coordinates $(\alpha, \alpha)$, and likewise for the columns, which could be equivalently formulated as re-scaling the model parameters by this factor. The same plots using diffusion models and of the raw values $F_{\alpha\beta}$ (without row and column-wise normalisation) are presented in Appendix B.3, Figures 14 & 15.

where $F_{\boldsymbol{\theta}}$ is called the *Fisher Information Matrix (FIM)*. Intuitively, the FIM re-scales the gradients to give more equal weighting to the parameters which typically have smaller gradients, and thus the Fisher information metric accounts for and is independent of how the parameters are scaled. This in theory prevents a dependence on representation in the gradient space.

The value $\|\nabla_{\boldsymbol{\theta}} l(\boldsymbol{x})\|^2_{FIM}$ is called the score statistic, which Rao (1948) showed to follow a $\chi^2$ distribution with $|\boldsymbol{\theta}|$ degrees of freedom, assuming the model parameters $\boldsymbol{\theta}$ are maximum likelihood estimates. Thus the score statistic can be used with a statistical test known as the *score test* Radhakrishna Rao (1948), as was used by Choi et al. (2021); Bergamin et al. (2022). Deep generative models with a likelihood-based objective perform a form of approximate maximum-likelihood estimation.

### 3.3 Approximating the Fisher Information Matrix

In practise, the full FIM has $P \times P$ entries, where $P = |\boldsymbol{\theta}|$ is the number of parameters of the deep generative model. This means that is too large to store in memory, and would furthermore be too expensive to compute and invert. For example our glow implementation has $P \approx 44$ million parameters, and thus the FIM would require $\approx 7,700$ terabytes to store using a `float32` representation. To develop a computable method, we therefore need to find a way to approximate the FIM. What would be a good choice for this approximation? – This problem is non-trivial due to its dimensionality. We start answering this question by computing the FIM in Eq. (2) restricted to a subset of parameters in two layers with parameters $\boldsymbol{\theta}_1, \boldsymbol{\theta}_2$ of a Glow Kingma & Dhariwal (2018) model trained on `CelebA`, using the Monte-Carlo (MC) approximation[2]

$$F_{\boldsymbol{\theta}_j} = E_{\boldsymbol{y} \sim p^{\boldsymbol{\theta}}}(\nabla_{\boldsymbol{\theta}_j} l(\boldsymbol{y}) \nabla_{\boldsymbol{\theta}_j} l(\boldsymbol{y})^T) \approx \frac{1}{N} \sum_{i=1}^{N} \nabla_{\boldsymbol{\theta}_j} l(\boldsymbol{y}^{(i)}) \nabla_{\boldsymbol{\theta}_j} l(\boldsymbol{y}^{(i)})^T, \qquad \boldsymbol{y}^{(i)} \sim p^{\boldsymbol{\theta}}, \qquad (3)$$

---

[2]Note that our choice to use a MC approximation is just for the sake of being able to compute $F_{\boldsymbol{\theta}}$; we here do not make any further (and more principled) approximations or assumptions.

where $\nabla_{\boldsymbol{\theta}_j}$ refers to taking the gradient with respect to the parameters $\boldsymbol{\theta}_j$ in layer $j$ and $\boldsymbol{y}^{(i)}$ are samples drawn from the generative model $p^{\boldsymbol{\theta}}$. This computation is infeasible for larger layers of the network which may be highly informative for OOD detection, demonstrating the need for a more principled approximation technique.

In Fig. 5 we illustrate the resulting (restricted) FIM estimate for two layers with $N = 1024$. Further layers of this and other models, and when trained on other datasets are presented in Appendix B.3, Figures 5 - 15. We observe an interesting pattern of *diagonal dominance*: The diagonal elements are significantly larger in absolute value, on average approximately five times the size of the off-diagonal elements. Hence, a seemingly 'crude', yet as turns out highly efficient approximation of the layer-wise FIM is to approximate as a multiple of the identity matrix, which corresponds to computing the *layer-wise $L^2$ norm* of the gradient. This reduces the cost from inverting an arbitrary $P \times P$ with potentially large P to approximating one value for each of the layers $\boldsymbol{\theta}_1, \boldsymbol{\theta}_2 \dots \boldsymbol{\theta}_J$ of the model.

### 3.4 A method for exploiting layer-wise gradients

We are now interested in operationalising our observations so far into an algorithm that can be used in practice.

In addition to the diagonal dominance phenomenon enabling a layer-wise approximation via a diagonal matrix, recall that layer-wise gradients contain more information than the overall gradient norm as the scale of the gradient norms differs by orders of magnitudes. We are therefore motivated to consider each layer $\boldsymbol{\theta}_1, \boldsymbol{\theta}_2 \dots \boldsymbol{\theta}_J$ in our model separately and combine the results as an OOD score in the second step. Specifically, if we select a layer $\boldsymbol{\theta}_j$ we can consider a restricted model where only the parameters in this layer are variable, and the other layers are frozen. We can approximate the score statistic (2) on this restricted model, whose parameters are more homogeneous in nature. In practice, we take the score vector for a layer $\nabla_{\boldsymbol{\theta}_j} l(\boldsymbol{x})$ and attempt to approximate $\left\| \nabla_{\boldsymbol{\theta}_j} l(\boldsymbol{x}) \right\|_{FIM}^2$, which should follow a $\chi^2$ test with $|\boldsymbol{\theta}_j|$ degrees of freedom for in-distribution data. As discussed in §3.3, we approximate the FIM restricted to this layer as a multiple of the identity. For a batch of $B$ (possibly equal to 1) of data points $\boldsymbol{x}_{1:B}$ we define features $f_{\boldsymbol{\theta}_j}$, which via our identity-matrix approximation should be proportional to $\left\| \nabla_{\boldsymbol{\theta}_j} l(\boldsymbol{x}_{1:B}) \right\|_{FIM}^2$, by

$$f_{\boldsymbol{\theta}_j}(\boldsymbol{x}_{1:B}) = \left\| \nabla_{\boldsymbol{\theta}_j} \left( \sum_{b=1}^{B} l(\boldsymbol{x}_b)) \right) \right\|_2^2. \tag{4}$$

Given that these layer-wise $L^2$ norms $f_{\boldsymbol{\theta}_j}$ should be proportional to a $\chi^2$ distributed variable with a large degree of freedom, we expect $\log f_{\boldsymbol{\theta}_j}$ to be normal-distributed Bartlett & Kendall (1946). In Fig. 4 (further results in Appendix B) we observe a good fit of $\log f_{\boldsymbol{\theta}_j}$ to a Normal distribution, empirically validating this holds in spite of our approximations.

This gives rise to a natural method of combining the layer-wise $L^2$ norms: we simply fit Normal distributions to each log-feature $\log f_{\boldsymbol{\theta}_j}$ independently, and then use the joint density as an "in-distribution" score. Algorithms 1 to 3 summarise our proposed method. As with other unsupervised OOD detection methods (Nalisnick et al., 2019b), we assume the existence of a small fit set of data held-out during training to accurately fit to each feature $f_j$. Note that in practise our method is very straight-forward to implement, requiring only a few lines of PyTorch code. Many other methods could possibly be constructed from our theoretical and empirical insights, and we will discuss potential future work in §6.

In Appendix B.5 we observe a mild performance improvement, uniformly across datasets, with the joint density approach in comparison to using Fisher's method Fisher (1938) when combining these statistics using z-scores. We hypothesise that this could be due to the density being more robust to correlation between adjacent layers as noted in Fig. 6. Our presented method does not enjoy the full invariance under rescaling of the model parameters as the true score statistic (see §3.2). However, in Appendix A.4 we show that it does satisfy invariance when rescaling each layer individually, justifying our use of the density in this setting. Our method satisfies the desiderata of the principle of invariance (see §4), is hyperparameter-free, and is applicable to any data modality and model with a differentiable estimate of the log-likelihood.

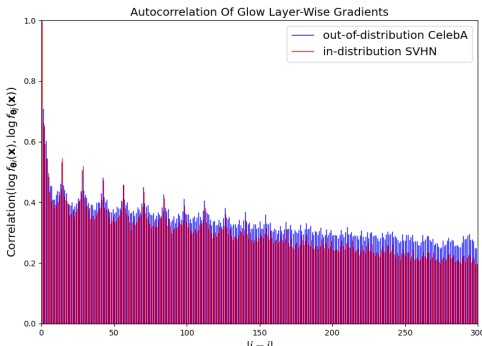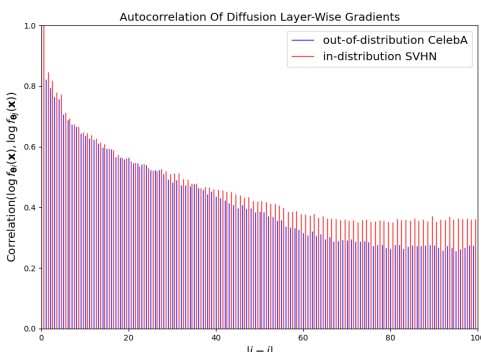

Figure 6: *The $L^2$-norms of layer-wise gradients have little correlation.* We select layers with parameters $\boldsymbol{\theta}_i, \boldsymbol{\theta}_j$ and measure the correlation of the logarithm gradient $L^2$-norms $\log f_{\boldsymbol{\theta}_i}(\boldsymbol{x})$. Binning these correlations by the distance between the layers $|i - j|$ and averaging across correlations of this distance gives the above plot. We note that there is a strong correlation in $L^2$-norm between adjacent layers, but that this correlation quickly decays for both in-distribution and out-of-distribution data. We hypothesise that this enables our approximation of the FIM which assumes independence across layers to provide good performance.

---

**Algorithm 1** Algorithm for computing features

**Require:** Deep generative model $M$, with parameters $\boldsymbol{\theta}_1, \boldsymbol{\theta}_2, \ldots \boldsymbol{\theta}_J$ in each of its $J$ layers.

    **function** GRADIENT FEATURES($\boldsymbol{x}_1 \ldots \boldsymbol{x}_B$)
        **for** $\boldsymbol{x_b}$ in batch **do**
            $l(\boldsymbol{x}_b) \leftarrow M(\boldsymbol{x}_b)$                                        ▷ Compute the log-likelihood
        **end for**
        $v_{\boldsymbol{\theta}} \leftarrow \nabla_{\boldsymbol{\theta}}(l(\boldsymbol{x}_1) + \cdots + l(\boldsymbol{x}_B))$           ▷ Compute the gradient via backpropagation
        **for** $j \leftarrow 1 \ldots J$ in layers **do**
            $f_j \leftarrow \left\| v_{\boldsymbol{\theta}_j} \right\|_2^2$                              ▷ Store the layer-wise $L^2$ norms
        **end for**
    **end function**

---

**Algorithm 2** Algorithm for training models

**Require:** train dataset and held-out fit dataset
    Train a deep generative model $M$, with parameters $\boldsymbol{\theta}_1, \boldsymbol{\theta}_2, \ldots \boldsymbol{\theta}_J$ in each of its $J$ layers.
    **for** Batch $\boldsymbol{x}_1^n \ldots \boldsymbol{x}_B^n$ in fit dataset **do**
        $f_1^n \ldots f_J^n \leftarrow$ GRADIENT FEATURES($\boldsymbol{x}_1^n \ldots \boldsymbol{x}_B^n$)
    **end for**
    **for** $j \leftarrow 1 \ldots J$ in layers **do**
        $\mu_j \leftarrow$ MEAN($\log f_j^1 \ldots \log f_j^N$)
        $\sigma_j^2 \leftarrow$ VARIANCE($\log f_j^1 \ldots \log f_j^N$)              ▷ Fit Gaussians to logarithmic features
    **end for**

---

**Algorithm 3** Algorithm for detecting OOD data

    Given new batch of samples $\boldsymbol{y}_1 \ldots \boldsymbol{y}_B$
    $\boldsymbol{f} \leftarrow$ GRADIENT FEATURES($\boldsymbol{y}_1 \ldots \boldsymbol{y}_B$)
    $S(\boldsymbol{y}_1 \ldots \boldsymbol{y}_B) = -\log \mathcal{N}(\log \boldsymbol{f}; \boldsymbol{\mu}; \mathrm{Diag}(\boldsymbol{\sigma}^2))$     ▷ Set OOD score to be the Gaussian negative log likelihood

### 3.5 Application to diffusion models

A denoising diffusion model Sohl-Dickstein et al. (2015a); Ho et al. (2020) uses a chain of latent variables $\{\boldsymbol{x}_n\}_{t=0}^{t=T}$ achieved by gradually adding noise (represented by the conditional distributions $q(\boldsymbol{x}_t|\boldsymbol{x}_{t-1})$) to the initial distribution of images $\boldsymbol{x} = \boldsymbol{x}_0$ until the final latent variable $\boldsymbol{x}_T$ is approximately Gaussian with mean zero and identity variance. The inverse process is then learned by a model approximating $p^{\boldsymbol{\theta}}(\boldsymbol{x}_{t-1}|\boldsymbol{x}_t)$. Diffusion models have gained in popularity due to their ability to produce samples with high visual fidelity and semantic coherence, making them a natural candidate for OOD detection Graham et al. (2023). Nonetheless, the full variational lower bound on the log-likelihood defined by Ho et al. (2020) is expensive to compute as it requires running $|T|$ inference steps on the network modelling $p^{\boldsymbol{\theta}}(\boldsymbol{x}_{t-1}|\boldsymbol{x}_t)$. For our setup, $|T| = 1000$ and running a full-forward/backward pass requires roughly 1 minute of compute per sample. Thus, we choose one component of the variational lower bound, namely the one-step log-likelihood,

$$l(\boldsymbol{x}) = l(\boldsymbol{x}_0) = \mathbb{E}_{\boldsymbol{x}_1 \sim q(\boldsymbol{x}_1|\boldsymbol{x}_0)} \log p^{\boldsymbol{\theta}}(\boldsymbol{x}_0|\boldsymbol{x}_1)$$

We refer to Appendix B.4 for computational details, noting that our implementation using one sample from $q(\boldsymbol{x}_1|\boldsymbol{x}_0)$ only requires 1 pass on of inference on $p^{\boldsymbol{\theta}}(\boldsymbol{x}_0|\boldsymbol{x}_1)$. Despite this value being very different to the intractable full log-likelihood $p^{\boldsymbol{\theta}}(\boldsymbol{x}_0)$, in Figure 2 we observe the same open problem and phenomenon as Nalisnick et al. (2019a) reported for the full likelihood estimates of other deep generative models. In Appendix B.4 we perform an ablation study on this one-step component of the variational lower bound used, finding the result that for both our method and the typicality method Nalisnick et al. (2019b), the components which are computed with less noise added to the image $\boldsymbol{x}_t$ used as input to the model $p^{\boldsymbol{\theta}}(\boldsymbol{x}_{t-1}|\boldsymbol{x}_t)$ are more informative for OOD detection, which could intuitively be understood as the noised image $\boldsymbol{x}_t$ itself being more informative in this regard.

## 4 Related work

In this section we review related work which uses gradients for unsupervised OOD detection.

In concurrent work to ours, Choi et al. (2021); Bergamin et al. (2022) each present an approximation to Rao's score test (Radhakrishna Rao, 1948). They independently approached the problem from the directions of training on the given sample of OOD data Xiao et al. (2020) and application of tests from classical statistics, respectively. These methods use approximations of the FIM from the field of optimization Amari (1998); Tieleman et al. (2012); Kingma & Ba (2015), whereas we use a simpler approximation tailored to the task of unsupervised OOD detection, and complement this with our empirical observations of the FIM in §3.3. Bergamin et al. (2022) compute a score test across the whole model by approximating the FIM as a diagonal, with elements $F_{\alpha\alpha} = (\partial_\alpha \log p^{\boldsymbol{\theta}}(\boldsymbol{x}))^2 + \epsilon$ for a small hyperparameter $\epsilon = 10^{-8}$, which which is used in optimization for its damping effect Martens (2020) and helps to mitigate numerical instabilities when dividing by $F_{\alpha\alpha}$. Our method differs in that it explicitly encodes the layer-homogeneity of the model (whereby parameters in the same layer have similar gradient sizes and perform similar functions in the network), and the predicted chi-square distribution of the score. We also note that layers which contain squared gradient values that are $<< 10^{-8}$ (see Appendix B.3 Figures 14 and 15) would have their information nullified without careful tuning of $\epsilon$, this can further be observed in 10 and 11 where there are entire layers which provide informativity for OOD detection and and have an $L^2$ norm of $< 10^{-8}$. Choi et al. (2021) split the problem of OOD detection layer-wise, but use the more complex EKFAC George et al. (2018) algorithm to account for dependencies between adjacent parameters. After some normalisation and additional processing steps, the authors compute the ROSE metric by taking the maximum feature over some pre-selected subset of layers. Our method differs as it uses a holistic score influenced by all the model layers.

Nguyen et al. (2019) are interested in using VAEs to detect anomalous web traffic in a semi-supervised setting, measuring the difference between a test gradient and labelled anomalous gradients. Our method differs in that does not require anomalous examples. Kwon et al. (2020) computes a cosine similarity between the gradients in the decoder of a VAE and the average gradients observed during training as their OOD metric of choice. In this work, we advocate for using the *size* of the gradient vector rather than its angle as done in Kwon et al.

(2020): our intuition is that, for a well-trained model evaluated on in-distribution data, we are close to a local minimum where the gradient is flat and the variance of the angle of the gradient vector is high. In particular, when averaging over samples from the model, we have that $\mathbb{E}_{\boldsymbol{x} \sim p^{\boldsymbol{\theta}}} \nabla_{\boldsymbol{\theta}}(\log p^{\boldsymbol{\theta}})(\boldsymbol{x}) = \boldsymbol{0}$ as a distribution minimises its own cross-entropy. In their supplementary, Nalisnick et al. (2019b) note that the Maximum Mean and Kernelized Stein Discrepancy tests they use to benchmark their typicality test only achieve good performance when using the inner product of the parameter gradients $k(x_i, x_j) = \nabla_\theta l(x_i)^T \nabla_\theta l(x_j)$, leading them to use the inner product with respect to the data in their experiments $k(x_i, x_j) = \nabla_{\boldsymbol{x}} l(x_i)^T \nabla_{\boldsymbol{x}} l(x_j)$. Our method differs from using inner product with respect to the parameters in that it allows for information to be used from all the layers, rather than a few dominant layers, as we discuss in §3.1.

To the best of our knowledge, no previous work has connected these works bar Bergamin et al. (2022)'s citation of Choi et al. (2021). The theoretical grounding which we provide in Proposition 1 §A.1 may explain why multiple other authors have independently found efficacy in unsupervised OOD detection with gradient information.

For completeness, in Appendix C we review previous work on the use of gradients of classifiers for the task of supervised OOD detection.

## 5 Experimental benchmark

| test ↓ train → | | SVHN | CelebA | GTSRB | CIFAR-10 | ImageNet32 |
|---|---|---|---|---|---|---|
| typicality (B = 1) | SVHN | - | 0.8735 | 0.3469 | 0.8599 | **0.8915** |
| | CelebA | **0.9989** | - | 0.6506 | 0.3680 | 0.2857 |
| | GTSRB | 0.9261 | 0.8201 | - | 0.6708 | 0.5548 |
| | CIFAR-10 | **0.9829** | 0.7733 | 0.6423 | - | 0.4147 |
| | ImageNet32 | 0.9952 | 0.9251 | 0.8057 | 0.7249 | - |
| ours (B = 1) | SVHN | - | **0.9880** | **0.9858** | **0.8747** | 0.8010 |
| | CelebA | 0.9823 | - | **0.9262** | **0.5155** | **0.2997** |
| | GTSRB | **0.9537** | **1.0000** | - | **0.7546** | **0.9967** |
| | CIFAR-10 | 0.9658 | **0.9462** | **0.9126** | - | **0.4377** |
| | ImageNet32 | **0.9976** | **0.9876** | **0.9683** | **0.7375** | - |
| typicality (B = 5) | SVHN | - | 0.9899 | 0.6119 | 0.9961 | **0.9983** |
| | CelebA | **1.0000** | - | 0.9786 | 0.4737 | 0.4293 |
| | GTSRB | **0.9997** | 0.8987 | - | 0.6639 | 0.6138 |
| | CIFAR-10 | **1.0000** | 0.9082 | 0.9613 | - | 0.4894 |
| | ImageNet32 | **1.0000** | 0.9974 | 0.9954 | 0.9013 | - |
| ours (B = 5) | SVHN | - | **0.9997** | **1.0000** | **0.9989** | 0.9976 |
| | CelebA | 0.9997 | - | **1.0000** | **0.9525** | **0.8514** |
| | GTSRB | 0.9996 | **0.9999** | - | **0.9596** | **0.9999** |
| | CIFAR-10 | 0.9992 | **0.9970** | **1.0000** | - | **0.6712** |
| | ImageNet32 | **1.0000** | **0.9995** | **1.0000** | **0.9480** | - |

Table 1: Comparison of the AUROC values (higher is better) of our method to the typicality test Nalisnick et al. (2019b) for batch sizes $B = 1, 5$. We train Glow Kingma & Dhariwal (2018) models on five natural image datasets (columns) and evaluate the ability of the model-method combination to reject the other datasets (rows). Bold indicates the element-wise higher value comparing both methods.

In this section, we benchmark our OOD detection method against the typicality test. We postpone detailed description of our datasets and models to Appendix D.

We follow the consensus of previous literature Nalisnick et al. (2019a) to evaluate our method on distribution pairs: training a generative image models on one image distribution and testing against a surrogate out-distribution. We choose five natural image datasets SVHN, CelebA, GTSRB, CIFAR-10 and ImageNet32 used in

previous literature Serrà et al. (2020) and evaluate on all dataset pairings. To the best of our knowledge this makes our evaluation more extensive than any previously published work in unsupervised OOD detection, a field where rigorous evaluation is especially important as erroneously high performance can be achieved by selective reporting of or fine-tuning hyperparameters to certain out-distributions.

In Tables 1 & 2 we compare our method against the typicality test Nalisnick et al. (2019b) using the Area Under Receiver Operating Curve (AUROC) statistic on both single sample ($B = 1$) batch size ($B = 5$) OOD detection. We choose typicality as it is, to the best of our knowledge, the most performant method which is both model-agnostic and hyper parameter free. Performance of unsupervised OOD detection can vary greatly depending on the model and even the image resizing algorithm applied make the inputs of uniform size Bergamin et al. (2022). To mitigate this problem we directly compare to our implmentation of Nalisnick et al. (2019b) using the same models and dataset implementations where they exist.

| $test \downarrow train \rightarrow$ | | SVHN | CelebA | GTSRB | CIFAR-10 | ImageNet32 |
|---|---|---|---|---|---|---|
| typicality ($B = 1$) | SVHN | - | 0.9357 | 0.4661 | **0.9007** | **0.8777** |
| | CelebA | **0.9990** | - | 0.3860 | 0.3409 | 0.2837 |
| | GTSRB | **0.9335** | 0.8197 | - | **0.6981** | **0.5624** |
| | CIFAR-10 | **0.9920** | 0.6968 | 0.4855 | - | 0.4142 |
| | ImageNet32 | **0.9986** | 0.8759 | **0.6759** | **0.7443** | - |
| ours ($B = 1$) | SVHN | - | **0.9903** | **0.8526** | 0.5574 | 0.6214 |
| | CelebA | 0.9551 | - | **0.5466** | **0.5655** | **0.3571** |
| | GTSRB | 0.8691 | **0.9684** | - | 0.5622 | 0.5530 |
| | CIFAR-10 | 0.9535 | **0.9639** | **0.5786** | - | **0.4710** |
| | ImageNet32 | 0.9363 | **0.9818** | 0.6651 | 0.5763 | - |
| typicality ($B = 5$) | SVHN | - | 0.9978 | 0.7943 | **0.9975** | **0.9961** |
| | CelebA | **1.0000** | - | 0.7642 | 0.3156 | 0.3621 |
| | GTSRB | **0.9998** | 0.8336 | - | 0.6809 | 0.5765 |
| | CIFAR-10 | **1.0000** | 0.7808 | **0.8332** | - | 0.4488 |
| | ImageNet32 | **1.0000** | 0.9866 | **0.9675** | **0.9266** | - |
| ours ($B = 5$) | SVHN | - | **1.0000** | **0.9970** | 0.8457 | 0.9561 |
| | CelebA | 0.9908 | - | **0.8552** | **0.7734** | **0.4202** |
| | GTSRB | 0.9716 | **0.9997** | - | **0.7325** | **0.9007** |
| | CIFAR-10 | 0.9895 | **0.9992** | 0.8104 | - | **0.5733** |
| | ImageNet32 | 0.9862 | **1.0000** | 0.9309 | 0.8532 | - |

Table 2: *Diffusion models* Comparison of the AUROC values (larger values are better) of our method to the typicality test Nalisnick et al. (2019b) for batch sizes $B = 1, 5$. We train Diffusion Ho et al. (2020) models on five natural image datasets (as columns) and evaluate the ability of the model-method combination to reject the other datasets (as rows). Bold indicates the element-wise higher value comparing both methods.

For Glow models (Table 1) our method outperforms typicality on most dataset pairings, whereas for diffusion models (Table 2) neither method dominates, although our method achieves higher average AUROC. We hypothesise that the comparative advantage our method enjoys for Glow models could be related to the model having more layers (1353 vs. 276) leading to more gradient features being available. In Appendix E table 7 we note poor performance for both methods when applied to a VAE model with poor sample quality, indicating that how well the model captures the dataset is the main factor driving performance of the downstream OOD detection method.

Single sample ($B = 1$) performance for both methods was lower for models trained on the semantically diverse datasets CIFAR-10 and ImageNet32. We mainly choose to include these datasets as in-distributions for consistency with prior work, and we would like to question the implicit assumption made by these prior works that an unsupervised method trained on these datasets *should* consistently reject images from other

natural image datasets. There is no obvious, meaningful semantic boundary to distinguish a natural image from `CIFAR-10` and `ImageNet32`, and thus it is not clear a even a human would outperform a random baseline.

As noted in §4 our method is similar to those presented in concurrent works Choi et al. (2021); Bergamin et al. (2022), we choose to use our method as a representative of this class of methods so that we may use our compute resources to robustly investigate their performance over such a wide range of distribution pairings and models. We make no claim of superior performance over these methods.

# 6 Conclusion

We analysed an approximation to the Fisher information metric for OOD detection. Our work has two key limitations: First, while we have provided the most extensive empirical benchmark of deep generative models, OOD and in-distribution datasets, datasets beyond images and for instance large language models should be tested. Second, while we focused on comparing it to the best performing, model-agnostic, hyperparameter-free OOD method, further empirical benchmarking against other methods should be conducted. Future work should investigate other, potentially more computationally expensive methods for approximating the Fisher information metric and its use in OOD detection.

## Acknowledgements

Sam Dauncey acknowledges support from the University of Oxford Statistics Summer Research Internship programme.

Chris Holmes acknowledges support from the Medical Research Council Programme Leaders award MC_UP_A390_1107, The Alan Turing Institute, Health Data Research, U.K., and the U.K. Engineering and Physical Sciences Research Council through the Bayes4Health programme grant.

Christopher Williams acknowledges support from the Defence Science and Technology (DST) Group and from a ESPRC DTP Studentship.

Fabian Falck acknowledges the receipt of studentship awards from the Health Data Research UK-The Alan Turing Institute Wellcome PhD Programme (Grant Ref: 218529/Z/19/Z), and the Enrichment Scheme of The Alan Turing Institute under the EPSRC Grant EP/N510129/1.

We thank Joost van Amersfoort for his code repository and useful discussions.

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

# Appendix for Approximations to the Fisher Information Metric of Deep Generative Models for Out-Of-Distribution Detection

## A   Proofs and additional theoretical results

### A.1   Proof of proposition 1

**Proposition 1.**   Let $p_{\mathcal{X}}^{\boldsymbol{\theta}}(\boldsymbol{x})$ and $p_{\mathcal{T}}^{\boldsymbol{\theta}}(\boldsymbol{t})$ be two probability density functions corresponding to the same model distribution $p^{\boldsymbol{\theta}}$ being represented on two different measure spaces $\mathcal{X}$ and $\mathcal{T}$. Suppose these representations encode the same information, i.e. there exists a smooth, invertible reparameterization $T : \mathcal{X} \to \mathcal{T}$ such that for $\boldsymbol{x} \in \mathcal{X}$ and $\boldsymbol{t} \in \mathcal{T}$ representing the same point we have $T(\boldsymbol{x}) = \boldsymbol{t}$. Then, the gradient vector $\nabla_{\boldsymbol{\theta}}(\log p^{\boldsymbol{\theta}})$ is invariant to the choice of representation, and in particular, $\nabla_{\boldsymbol{\theta}}(\log p_{\mathcal{T}}^{\boldsymbol{\theta}})(\boldsymbol{t}) = \nabla_{\boldsymbol{\theta}}(\log p_{\mathcal{X}}^{\boldsymbol{\theta}})(\boldsymbol{x})$.

**Proof.**   Via the change-of-variables formula, we obtain

$$p_{\mathcal{T}}^{\boldsymbol{\theta}}(\boldsymbol{t}) = p_{\mathcal{X}}^{\boldsymbol{\theta}}(\boldsymbol{x}) \left| \frac{\partial T^{-1}}{\partial \boldsymbol{x}} \right|.$$

Applying the logarithm on both sides provides

$$\log p_{\mathcal{T}}^{\boldsymbol{\theta}}(\boldsymbol{t}) = \log p_{\mathcal{X}}^{\boldsymbol{\theta}}(\boldsymbol{x}) + \log \left| \frac{\partial T^{-1}}{\partial \boldsymbol{x}} \right|,$$

and hence $\nabla_{\boldsymbol{\theta}}(\log p_{\mathcal{T}}^{\boldsymbol{\theta}})(\boldsymbol{t}) = \nabla_{\boldsymbol{\theta}}(\log p_{\mathcal{X}}^{\boldsymbol{\theta}})(\boldsymbol{x})$ as required.

The smoothness assumption could be relaxed by considering the pull-back measure $\mathbb{P}_{\mathcal{X}}^{\boldsymbol{\theta}} \circ T^{-1} = \mathbb{P}_{\mathcal{T}}^{\boldsymbol{\theta}}$ and the corresponding change-of-variables formula for Radon-Nikodym derivatives, however we omit this for brevity and relevance. This result also trivially extends to the likelihood proxy we use for diffusion models $\log p^{\boldsymbol{\theta}}(\boldsymbol{x}_0|\boldsymbol{x}_1)$.

### A.2   Representation-Invariance of variational lower-bound gradients

Assume the same setup as in A.1, but this time with a variational Bayesian method such as a Variational AutoEncoder Kingma & Welling (2014) with latent variable given by $\boldsymbol{z}$, decoder probability density $p_{\mathcal{X}}^{\boldsymbol{\theta}}(\boldsymbol{x}|\boldsymbol{z})$ and encoder probability density $q^{\phi}(\boldsymbol{z}|\boldsymbol{x})$, noting that the decoder probability density is that which depends on $\mathcal{X}$. The Evidence Lower Bound on the log-likelihood $p_{\mathcal{X}}^{\boldsymbol{\theta}}(\boldsymbol{x})$ is given by

$$ELBO_{\mathcal{X}}^{\boldsymbol{\theta},\phi}(\boldsymbol{x}) = \mathbb{E}_{\boldsymbol{z} \sim q^{\phi}(\boldsymbol{z}|\boldsymbol{x})} \left( \log \frac{p_{\mathcal{X}}^{\boldsymbol{\theta}}(\boldsymbol{x}, \boldsymbol{z})}{q^{\phi}(\boldsymbol{z}|\boldsymbol{x})} \right).$$

We can then state a similar representation invariance for the ELBO.

**Proposition 2.** Let $ELBO_{\mathcal{X}}^{\boldsymbol{\theta},\phi}$ be the ELBO of a VAE, and let $ELBO_{\mathcal{T}}^{\boldsymbol{\theta},\phi}(\boldsymbol{t})$ be the ELBO under a change of variables with invertible mapping $T : \mathcal{X} \to \mathcal{T}$, corresponding to two sets $\mathcal{X}$ and $\mathcal{T}$. Then, the gradient $\nabla_{\boldsymbol{\theta}}(ELBO_{\mathcal{T}}^{\boldsymbol{\theta},\phi})(\boldsymbol{t})$ is invariant to $T$.

**Proof.**   Noting that     $p_{\mathcal{T}}^{\boldsymbol{\theta}}(\boldsymbol{t}, \boldsymbol{z}) = p_{\mathcal{X}}^{\boldsymbol{\theta}}(\boldsymbol{x}, \boldsymbol{z}) \left| \frac{\partial T^{-1}}{\partial \boldsymbol{x}} \right|$     while     $q^{\phi}(\boldsymbol{z}|\boldsymbol{t}) = q^{\phi}(\boldsymbol{z}|\boldsymbol{x})$     [3] gives that:

$$ELBO_{\mathcal{T}}^{\boldsymbol{\theta},\phi}(\boldsymbol{t}) = ELBO_{\mathcal{X}}^{\boldsymbol{\theta},\phi}(\boldsymbol{x}) + \log \left| \frac{\partial T^{-1}}{\partial \boldsymbol{x}} \right|$$

and taking the gradient wrt $\boldsymbol{\theta}$ and $\phi$ gives the result that the gradient of the ELBO with respect to the VAE's parameters is representation-invariant.

---

[3]To those familiar with the Borel–Kolmogorov paradox this condition may seem non-obvious, but we can derive it from the fact that $T$ does not require input from $\boldsymbol{z}$, and thus $q^{\phi}(\boldsymbol{z}|\boldsymbol{t}) = \frac{q^{\phi}(\boldsymbol{z},\boldsymbol{t})}{q^{\phi}(\boldsymbol{t})} = \frac{q^{\phi}(\boldsymbol{z},\boldsymbol{x}) \left| \frac{\partial T^{-1}}{\partial \boldsymbol{x}} \right|}{q^{\phi}(\boldsymbol{x}) \left| \frac{\partial T^{-1}}{\partial \boldsymbol{x}} \right|} = q^{\phi}(\boldsymbol{z}|\boldsymbol{x})$

### A.3 Lebesgue measure of a set of bounded total variation

**Proposition 3.** For $\boldsymbol{x} \in \mathbb{R}^d$, define the total variation to be $TV(\boldsymbol{x}) = |x_1| + \sum_{i=2}^{d} |x_i - x_{i-1}|$. Let $E(\alpha)$ be the set of $d$-length arrays whose total variation is bounded by $\alpha$:

$$E(\alpha) = \{\boldsymbol{x} \in \mathbb{R}^d : TV(\boldsymbol{x}) < \alpha\}.$$

The Lebesgue measure of this set is given by $E(\alpha) = \frac{(2\alpha)^d}{\Gamma(d+1)}$.

**Proof.**

Consider the volume-preserving transformation $(x_1, x_2 \ldots x_d) \mapsto (x_1, t_2 \ldots t_d)$, where $t_i = x_i - x_{i-1}$. We thus see that the volume of $E(\alpha)$ is equivalent to the volume of the $d$-ball in the $\ell^1$-metric, with a standard result:

$$\mu(E(\alpha)) = \mu(\{(x_1, t_2 \ldots t_d) : |x_1| + |t_2| + \cdots + |t_d|\} < \alpha) = \frac{(2\alpha)^d}{\Gamma(d+1)}.$$

**Application to MNIST** We can naïvely apply this result to MNIST images $\boldsymbol{y} \in [0,1]^{28 \times 28}$ by setting $d = 28^2$ and drawing a snake pattern through our images as illustrated in Figure 7, setting $y_{ij} = x_{28(j-1)+(-1)^{j+1}(i-14)+14}$. Computing this numerically for the whole MNIST dataset, we see that $\alpha = 102.9$ is sufficiently large such that the whole MNIST dataset is contained in $E(\alpha)$, which we can compute has an approximate measure of $E(\alpha) \approx 10^{-116.76} \leq 10^{-116}$. Note that this is not the tightest bound one could give; for example vertical variations are neglected and membership in $E(\alpha)$ does not restrict $x_i$ from drifting outside the set $[0,1]$.

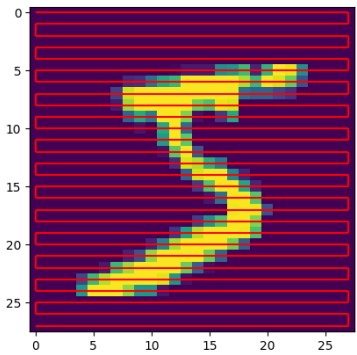

Figure 7: A visual illustration of the snake pattern used to unravel an MNIST image $\boldsymbol{y} \in [0,1]^{28 \times 28}$ into a string of values $\boldsymbol{x} \in [0,1]^{784}$

### A.4 (Weak) parameterisation in-variance of our method

Let $\Theta, \Phi$ be two parameter spaces of the same model $p$, linked by the smooth invertible reparameterisation $P : \Theta \to \Phi$, such that for $\boldsymbol{\phi} = P(\boldsymbol{\theta})$ we have $p^{\boldsymbol{\theta}} = p^{\boldsymbol{\phi}}$. In this setting, one can derive that the Fisher Information Metric Radhakrishna Rao (1948) is invariant under $P$, ie that for all $\boldsymbol{x}_1, \boldsymbol{x}_2 \in \mathcal{X}$ we have $\nabla_{\boldsymbol{\theta}} l(\boldsymbol{x}_1) F_{\boldsymbol{\theta}}^{-1} \nabla_{\boldsymbol{\theta}} l(\boldsymbol{x}_2)^T = \nabla_{\boldsymbol{\phi}} l(\boldsymbol{x}_1) F_{\boldsymbol{\phi}}^{-1} \nabla_{\boldsymbol{\phi}} l(\boldsymbol{x}_2)^T$ (see (2) for our notation). As we merely approximate the FIM in our method we cannot make the same guarantee for all $P$, we can however prove a similar result if $P$ linearly rescales the layers:

**Proposition 4.** As in §3.4 Let $\boldsymbol{\theta}_1, \boldsymbol{\theta}_2 \ldots \boldsymbol{\theta}_J$ be the layers of our model. $P : \Theta \to \Phi$ be a smooth invertible reparameterisation of our model which linearly rescales the layers, ie $P(\boldsymbol{\theta}_1, \boldsymbol{\theta}_2 \ldots \boldsymbol{\theta}_J) = d_1 \boldsymbol{\theta}_1, d_2 \boldsymbol{\theta}_2 \ldots d_J \boldsymbol{\theta}_J$ for some constants $d_1, d_2 \ldots d_J v \in \mathbb{R}$. Then, the resulting anomaly score of our method is invariant under $P$.

**Proof.**

Using the same notation as in §3.4, let $f_1^\Theta \ldots f_J^\Theta$ and $f_1^\Phi \ldots f_J^\Phi$ be our layer-wise gradient $L^2$ norm features under $\Theta$ and $\Phi$ respectively (see equation 4). Then, for all datapoints $\boldsymbol{x}$ and layers $j$ we have:

$$f_j^\Theta(\boldsymbol{x}) = \left\|\nabla_{\boldsymbol{\theta}_j} l(\boldsymbol{x})\right\|^2 = \left\|d_j \nabla_{\boldsymbol{\phi}_j} l(\boldsymbol{x})\right\|^2 = d_j^2 f_j^\Phi(\boldsymbol{x}). \tag{5}$$

Taking the logarithm and writing in vectorized form gives that:

$$\log \boldsymbol{f}^\Theta(\boldsymbol{x}) = 2\log \boldsymbol{d} + \log \boldsymbol{f}^\Phi(\boldsymbol{x}) \tag{6}$$

In particular, if we let $\boldsymbol{\mu}^\Theta, \boldsymbol{\mu}^\Phi$ and $\boldsymbol{\sigma}^{2\Theta}, \boldsymbol{\sigma}^{2\Phi}$ be the corresponding sample mean and variances for $\Theta$ and $\Phi$ in algorithm 3, we see that $\boldsymbol{\mu}^\Theta = 2\log \boldsymbol{d} + \boldsymbol{\mu}^\Phi$ and $\boldsymbol{\sigma}^{2\Theta} = \boldsymbol{\sigma}^{2\Phi} = \boldsymbol{\sigma}^2$. Hence via translation invariance of the normal distribution, our metric will be invariant under $P$.

### A.5 Comparison to classical invariance properties

In classical statistics, there is a separate and notion of invariance that is incompatible with that proposed by Le Lan & Dinh (2021). The setup proposed by Lehmann et al. (1986) is one in which we consider a group of transformations from the input space to itself $g : X \to X$ which are sufficiently narrow such that there is a corresponding group of transformations to the parameter space $\bar{g} : \theta \to \theta$ that counteract the effect of the transformation, formally defined by:

$$\mathbb{P}_\mathcal{X}^{\bar{g}\theta} = \mathbb{P}_\mathcal{X}^\theta \circ g^{-1}, \tag{7}$$

with the analogy to the change-of-variables formula being that:

$$p_\mathcal{X}^{\boldsymbol{\theta}}(g\boldsymbol{x}) = p_\mathcal{X}^{\bar{g}^{-1}\boldsymbol{\theta}}(\boldsymbol{x}) \left|\frac{\partial g\boldsymbol{x}}{\partial \boldsymbol{x}}\right|^{-1}. \tag{8}$$

One example of where this setup is applicable is applying dilations to an input space of a multivariate normal distribution, whereby any linear dilation of the input space can be counteracted by a dilation of the covariance matrix. This is not the case for the arbitrary transformations $f$ considered in Proposition 1 of Le Lan & Dinh (2021), which we cannot guarantee to be counteracted by some transformation of a generative model's parameters. Even the simple example of the non-linear RGB-HSV transformation we give in §2.3 can only approximately be counteracted by changing the generative model's parameters. In contrast, the setup proposed by Le Lan & Dinh (2021), implicitly considers transformations from arbitrary measure spaces $f : X \to f(X)$, and considers the pullback:

$$\mathbb{P}_{f(X)}^\theta = \mathbb{P}_X^\theta \circ f^{-1}. \tag{9}$$

The presence of the counteracting parameter transformation in equation (7) leads to a $\frac{\partial \bar{g}^{-1}\theta}{\partial \theta}$ term in the score vector (equation (6) of De Maio et al. (2010)):

$$\nabla_{\boldsymbol{\theta}}(\log p_\mathcal{X}^{\boldsymbol{\theta}})(g\boldsymbol{x}) = \nabla_{\boldsymbol{\theta}}(\log p_\mathcal{X}^{\bar{g}^{-1}\boldsymbol{\theta}})(\boldsymbol{x}) = \frac{\partial \bar{g}^{-1}\theta}{\partial \theta}\nabla_{\boldsymbol{\theta}}(\log p_\mathcal{X}^{\boldsymbol{\theta}})(\boldsymbol{x}). \tag{10}$$

Nonetheless, De Maio et al. (2010) derive that the score test statistic is invariant under this setup too. This is a consequence of the score test statistic being *both* invariant in the setup of Le Lan & Dinh (2021) *and* satisfying strong parameterisation invariance, which is algebraically expressed by the $\frac{\partial \bar{g}^{-1}\theta}{\partial \theta}$ term in eq (10) being cancelled in re-parameterisation of the FIM.

# B   Additional experimental details and results

## B.1   RGB-HSV representation dependence

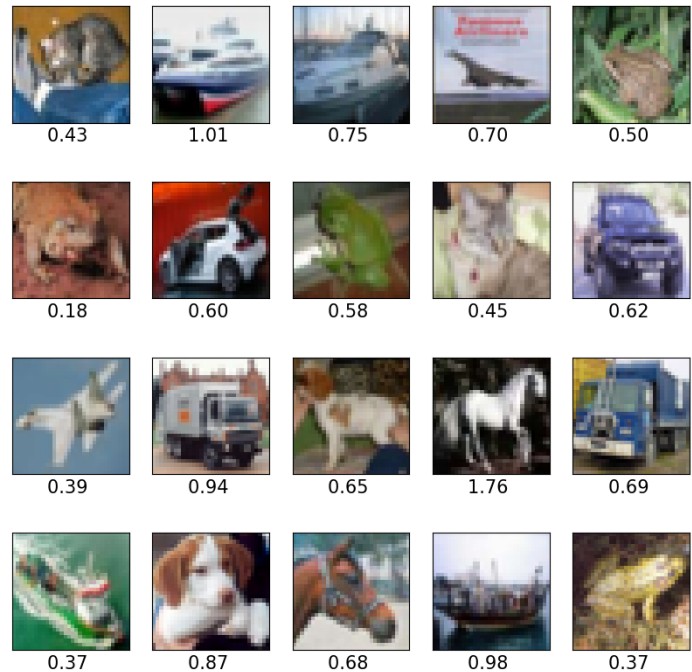

Figure 8: *The log-likelihood heavily depends on data representation* We extend Figure 3 to the first 20 examples of the `CIFAR10` dataset and their values of $\Delta_{BPD}^{RGB \to HSV}$ as defined in Eq. 11. We note values of $\Delta_{BPD}^{RGB \to HSV}$ between 0.18 to 1.76, indicating a large difference of the induced change in likelihoods.

Here we compute the change in Bits Per Dimension (BPD) $\Delta_{BPD}^{RGB \to HSV}$ for the first 20 samples of the `CIFAR10` test dataset, defined as:

$$\Delta_{BPD}^{RGB \to HSV} = \frac{1}{3 \times 32 \times 32} \log_2 \frac{d\mu_{HSV}}{d\mu_{RGB}} = \frac{\log_2 p_{RGB}(\mathbf{x}) - \log_2 p_{HSV}(\mathbf{x})}{3 \times 32 \times 32}, \tag{11}$$

where $\mu_{HSV}$ is the Lebesgue measure in HSV-space, $\mu_{RGB}$ is the Lebesgue measure in $RGB$-space, and $p_{HSV}$ and $p_{RGB}$ are corresponding probability density functions for any distribution defined over the set of images. We compute the Radon-Nikodym derivative $\frac{d\mu_{HSV}}{d\mu_{RGB}}$ in 11 by computing the pixel-wise Jacobian determinants of the RGB-HSV transformation $T^{RGB \to HSV} : \mathbb{R}^3 \to \mathbb{R}^3$. In order to make the comparison fair, we dequantaize pixel $x_{ij} \in \mathbb{R}^3$ add a small amount of normally distributed noise $\epsilon_{ij} \sim \mathcal{N}(\mathbf{0}, \boldsymbol{I}_{3 \times 3})$, ie we set $\tilde{x}_{ij} = x_{ij} + \frac{\epsilon_{ij}}{255}$. We then note that the full RGB-HSV transformation factors as a RGB-HSV transformation across the pixels, and thus its Jacobian determinant factors as:

$$\log \frac{d\mu_{HSV}}{d\mu_{RGB}} = \sum_{1 \leq i,j \leq 32} \log \left| \frac{\partial T}{\partial x_{ij}} \right|_{\tilde{x}_{ij}}.$$

## B.2 Replications of Fig. 4

In figures 9- 11 we provide robust replications of Figure 4 using randomly chosen layers. The layers are sorted with the right-hand layer being the "deepest" (ie. the closest to the latent variables). We observe that the gradients are more separated for models trained on the semantically distinct datasets SVHN, CelebA & GTSRB, mirroring the superior performance our method achieves in these cases.

Please note that large parts of the gradient distributions from the OOD datasets have been cropped out to keep the plots legible.

### B.2.1 Glow models

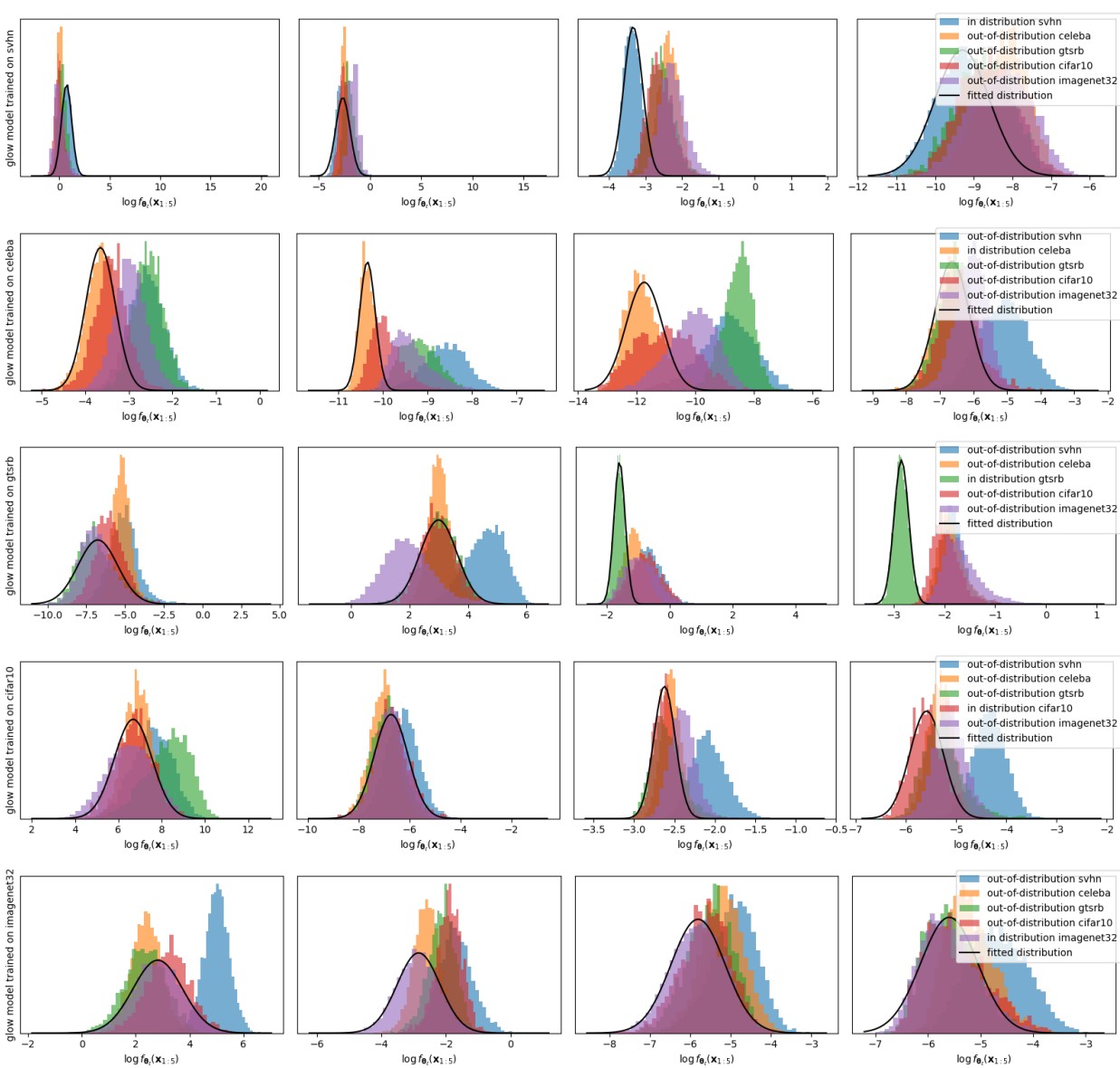

Figure 9: Replication of Fig. 4 for 4 randomly selected layers out of 1353 from Glow models trained on SVHN, CelebA, GTSRB, CIFAR-10 and ImageNet32 respectively

## B.2.2 Diffusion models

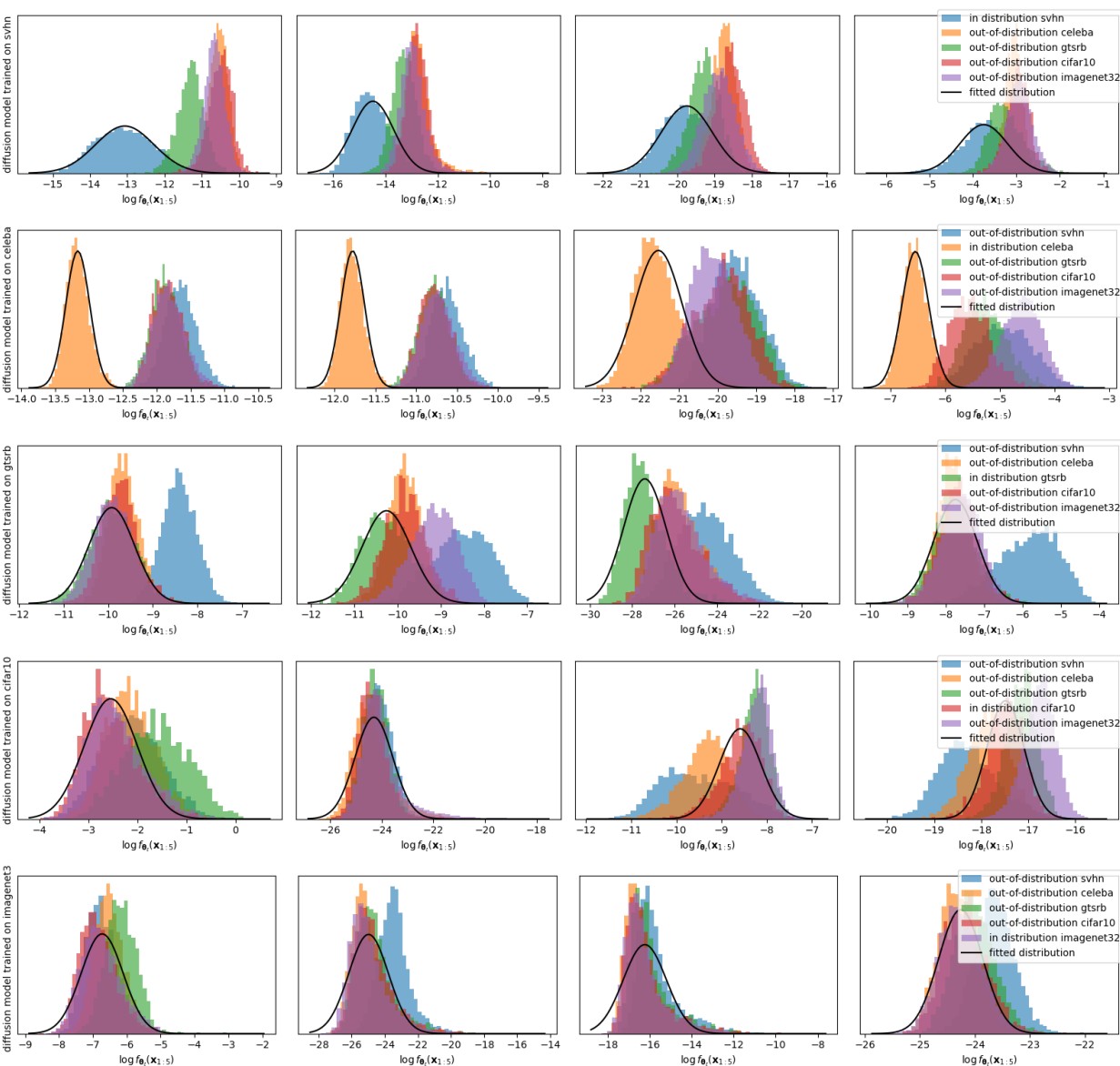

Figure 10: Replication of Fig. 4 for 4 randomly selected layers out of 276 from Diffusion models trained on SVHN, CelebA, GTSRB, CIFAR-10 and ImageNet32 respectively

### B.2.3 VAEs

We note generally less separation with our VAE models, mirroring the poorer performance we attain with them in appendix E

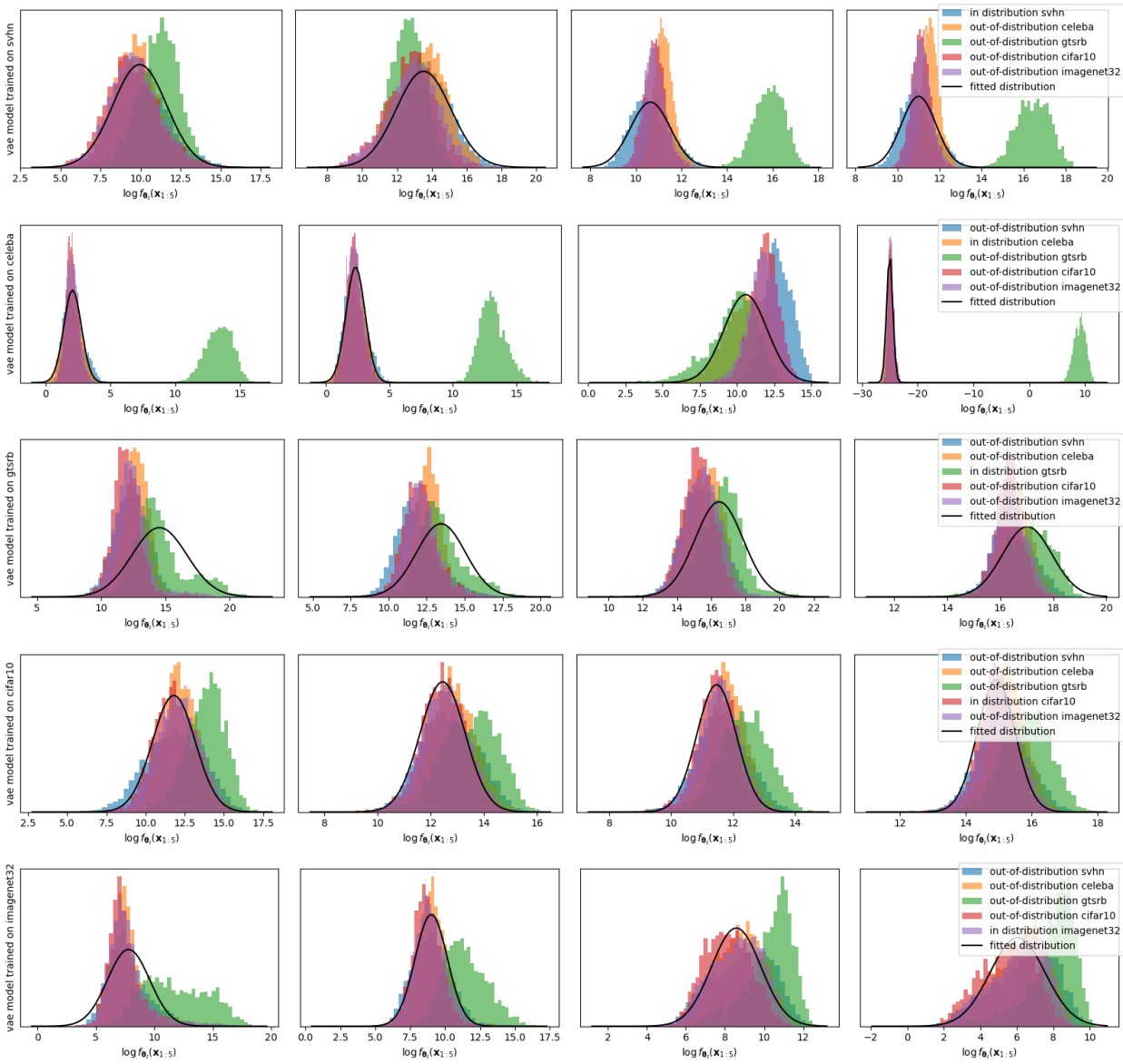

Figure 11: Replication of Fig. 4 for 4 randomly selected layers out of 48 from vae models trained on `SVHN`, `CelebA`, `GTSRB`, `CIFAR-10` and `ImageNet32` respectively

### B.3 Additional plots of the FIM

#### B.3.1 Windows into the FIM of a Glow model

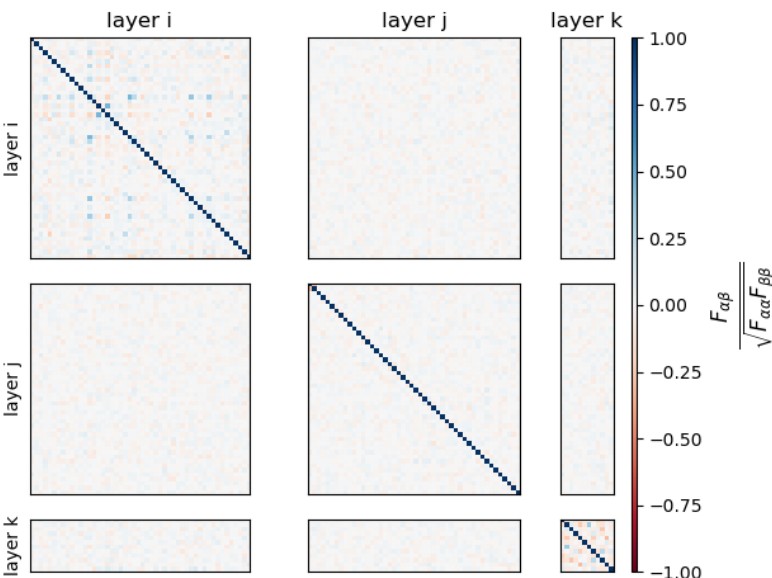

Figure 12: *The strong diagonal of the FIM for glow models* we replicate Fig. 5 using three more randomly selected layers from our Glow model trained on `CelebA`.

#### B.3.2 Windows into the FIM of a diffusion model

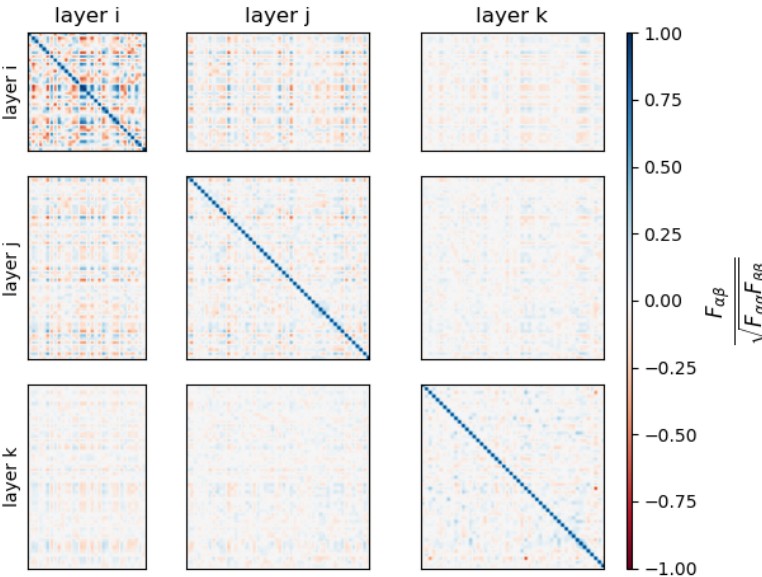

Figure 13: *The strong diagonal of the FIM for diffusion models.* We replicate Fig. 5 with a diffusion model. As before, we randomly select two layers $\theta_\ell$ from a diffusion model trained on `CelebA` and plot the an approximation of the FIM, this time using the gradients of the one-step log-likelihood. Note that the first layer selected has fewer than 50 weights, so we plot its entire layer-wise FIM. Again, we normalise the rows and columns by the diagonal values $\sqrt{F_{\alpha\alpha}}$ to enable cross-layer comparison.

### B.3.3 Raw, single-layer FIMs of Glow models

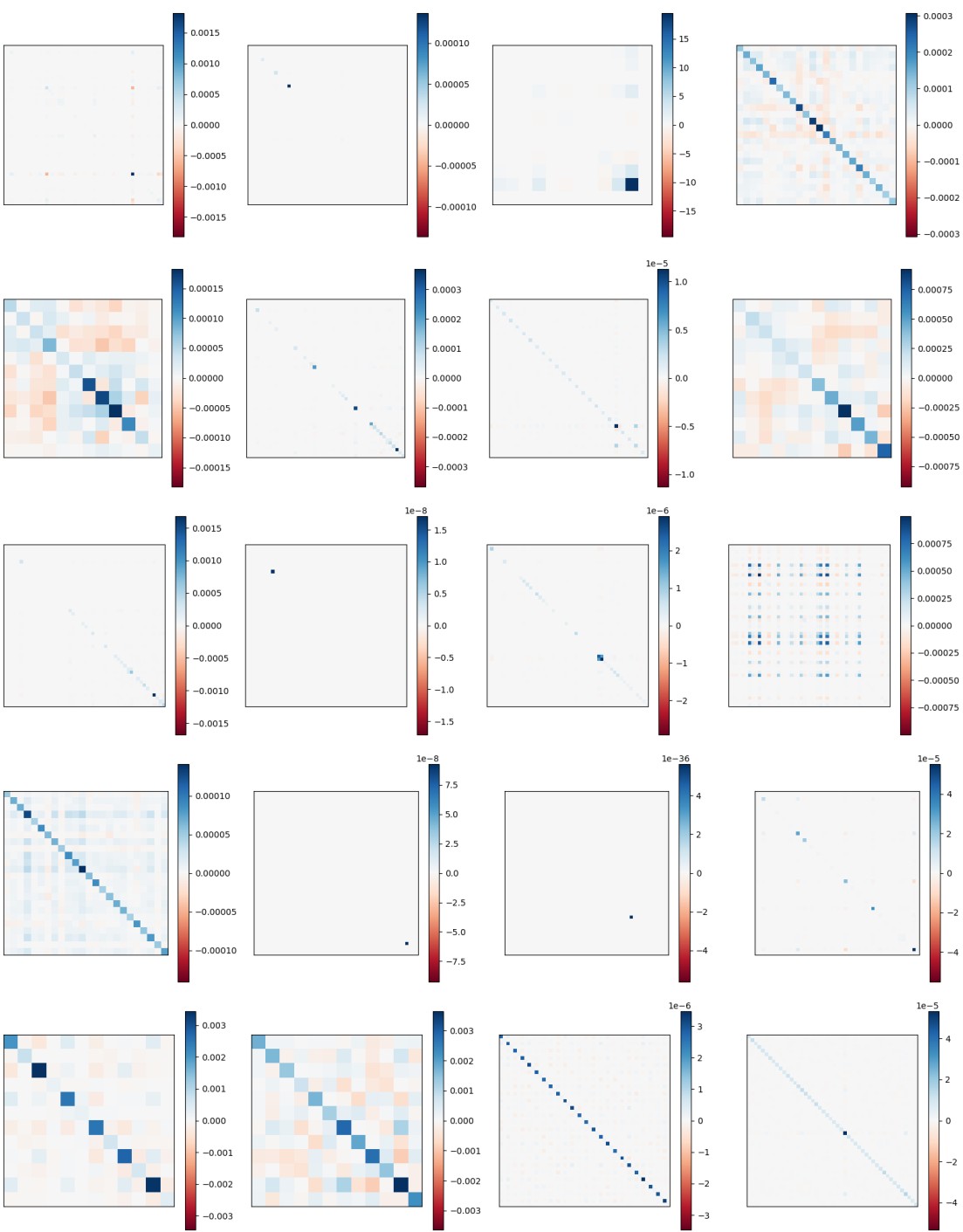

Figure 14: *Raw layer-wise FIMs for glow models.* For each row of plots, we randomly select 4 layers $\boldsymbol{\theta}_\ell$ from glow models trained on (going from top to bottom) `SVHN`, `CelebA`, `GTSRB`, `CIFAR-10` and `ImageNet32`. We then plot the raw FIM $F_{\alpha\beta}$ values for $\max(50, |\boldsymbol{\theta}|)$ weights in these layers, using a separate colorbar per layer to account for the fact that the absolute sizes of the FIM elements vary by orders of magnitudes from layer to layer.

### B.3.4 Raw, single-layer FIMs of diffusion models

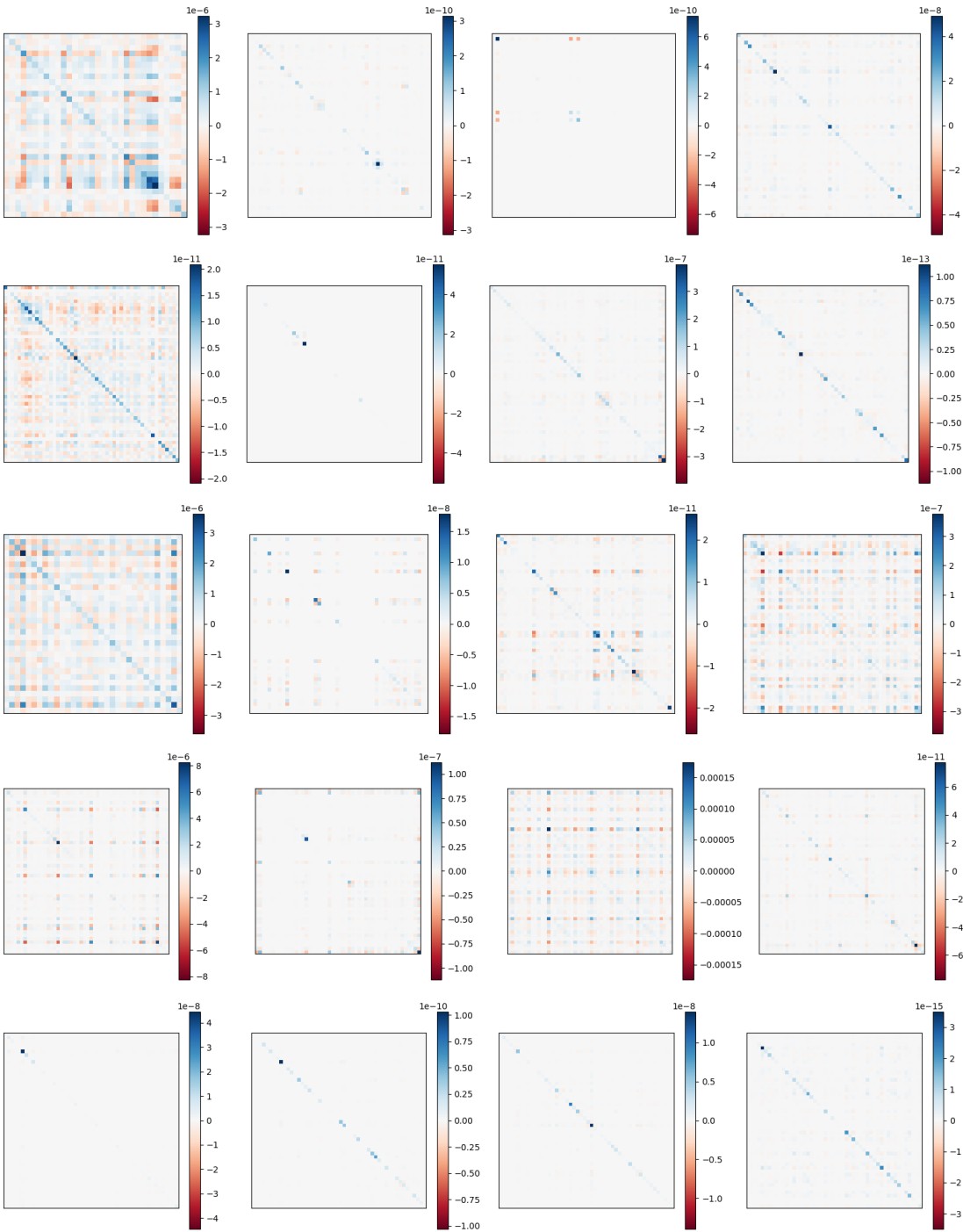

Figure 15: We replicate Figure 14 with a diffusion model (again using the gradient of the 1-timestep variational lower bound as a stand-in for the gradient of the log-likelihood), noting a qualitative difference in the appearance of the layers.

### B.4 Ablation study on likelihood proxies for diffusion models

In this section, we discuss the use of different parts of a diffusion variational lower bound for anomaly detection. Before doing so, for completeness we present a condensed version of the theory presented in Ho et al. (2020), using the same notation.

#### B.4.1 Derivation of diffusion process

Let our forward process be $q(\boldsymbol{x}_t|\boldsymbol{x}_{t-1})$, with prior of the true image distribution $q(\boldsymbol{x}_0)$ and our learned reverse process be $p^{\boldsymbol{\theta}}(\boldsymbol{x}_{t-1}|\boldsymbol{x}_t)$ with Gaussian prior $p(\boldsymbol{x}_T) \sim \mathcal{N}(\boldsymbol{0}, \boldsymbol{I})$. A variational lower bound on the model log-likelihood of can be derived Ho et al. (2020) as:

$$\mathbb{E}[-\log p^{\boldsymbol{\theta}}(\boldsymbol{x}_0)] \leq \mathbb{E}[L_T + \sum_{t>1} L_{t-1} + L_0] \tag{12}$$

$$\text{where} \quad L_t = \begin{cases} t = 0 & -\log p^{\boldsymbol{\theta}}(\boldsymbol{x}_0|\boldsymbol{x}_1) \\ 0 < t < T & KL(q(\boldsymbol{x}_t|\boldsymbol{x}_{t+1}, \boldsymbol{x}_0)||p^{\boldsymbol{\theta}}(\boldsymbol{x}_t|\boldsymbol{x}_{t+1})) \\ t = T & KL(q(\boldsymbol{x}_T|\boldsymbol{x_0})||p(\boldsymbol{x}_T)) \end{cases} \tag{13}$$

,

Ho et al. (2020) also derive that, if we parameterise our forward process as adding some normally distributed noise $\boldsymbol{\epsilon}$ to $\boldsymbol{x_{t-1}}$, and our reverse process as normally predicting this noise from $\boldsymbol{x}_t$ via a network $\boldsymbol{\epsilon}^{\boldsymbol{\theta}}(\boldsymbol{x_t}, t)$, for $0 \leq t < T$, $L_t$ can be computed as a squared error:

$$\mathbf{E}L_{t-1} = K_t \mathbf{E}_{\boldsymbol{x}_t \sim q(\boldsymbol{x}_0)} \left\| \boldsymbol{\epsilon} - \boldsymbol{\epsilon}^{\boldsymbol{\theta}}(\boldsymbol{x_t}, t) \right\|^2 + C_t. \tag{14}$$

Here $k_t$ and $C_t$ are constants independent of $\boldsymbol{x}_0, \boldsymbol{x}_1 \ldots \boldsymbol{x}_T$ and $\boldsymbol{\theta}$, which can thus be omitted from computations. To compute the expectation, we use one sample from the reverse process $\boldsymbol{x}_t \sim q(\boldsymbol{x}_0)$, motivated by our findings in section B.4.4 which show little to no performance gains from using five samples. Thus, we define our set of likelihood proxies as in (14) as $L_t(\boldsymbol{x}) = \left\| \boldsymbol{\epsilon} - \boldsymbol{\epsilon}^{\boldsymbol{\theta}}(\boldsymbol{x_{t+1}}, t+1) \right\|$ for a single sample of noise $\boldsymbol{\epsilon}$ from the reverse process $\boldsymbol{x}_{t+1} \sim q(\boldsymbol{x})$. Note that computing $L_t(\boldsymbol{x})$ requires only one pass through the network, making it very efficient to compute. In section B.4.3 we do an ablation study on the value of $t$ used, motivating our choice of $L_0$ in the application our method.

#### B.4.2 On Representation Dependence in Diffusion models

When considering Le Lan & Dinh (2021)'s results pertaining to representation dependence in the context of diffusion models, we arrive at the interesting question as to whether the choice of representation should be considered to affect the underlying distribution of the forward process $q$. Clearly the value we use in our method, $L_0 = p^{\boldsymbol{\theta}}(\boldsymbol{x}_0|\boldsymbol{x}_1)$, is representation-dependent. In the strict sense, the values $L_t$ for $t > 0$ aren't representation dependent, unless representation dependence is also considered to affect $q$, in which case this becomes more ambiguous. In figure 16 we report the negative result that the values of $L_t$ for $t > 0$ also follow the pattern from Nalisnick et al. (2019a), whereby structured OOD data has higher values for $L_t$. We defer further debate on this issue to future work.

#### B.4.3 Ablation study on the value of $t$ used for anomaly detection with diffusion models

In this section, we evaluate using different values of $t$ for the likelihood proxy $l(\boldsymbol{x}) = L_{t-1}(\boldsymbol{x})$ which we use as input for our anomaly detection method and typicality Nalisnick et al. (2019a). We summarise our results in Figure 17 by plotting the average AUROC acheived for each method across all 20 dataset pairings. In table 3 we provide more granular results with the AUROC for each pairing individually. We note the intuitive result that the performance of our method gradually decays as $t$ increases, corresponding to more noise being added to the sample fed into the network. Overall, the average performance of our method at $t = 1$ is higher than

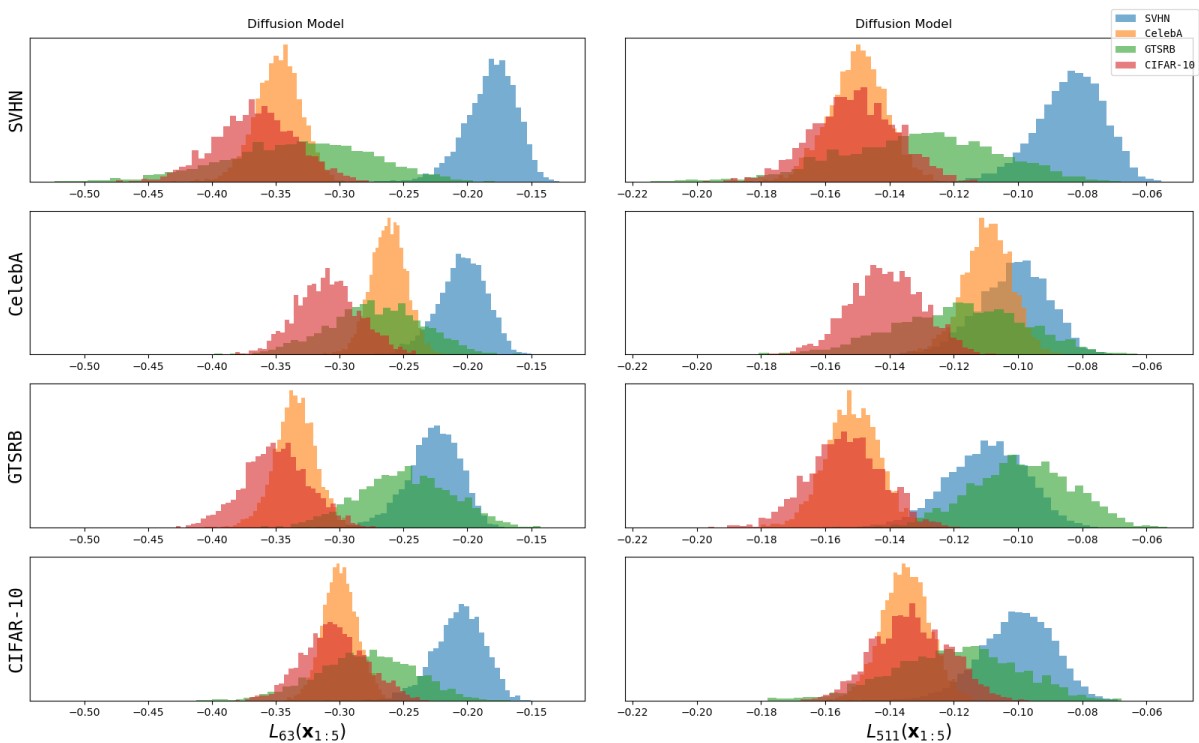

Figure 16: $L_{t-1}$ *follows the pattern from Nalisnick et al. (2019a) for a variety of t values.* We replicate figure 2 for $t = 64$ [Left] and $t = 512$ [Right] *using batch size $B = 5$* (we use this batch size for reasons of limited compute.)

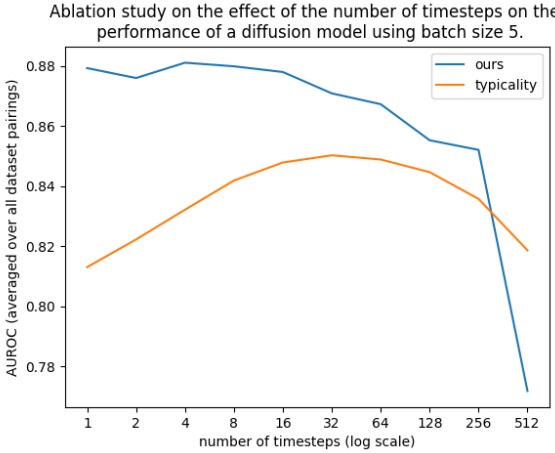

Figure 17: *For diffusion models, $L_{t-1}$ is most informative for anomaly detection for low values of t.* We compute the AUROC values for all in/out dataset distribution pairings using $t = 2^n$ for $n = 0, 1 \ldots 9$ and batch size $B = 5$ (for reasons of limited compute).

that of typicality which achieves its maximum performance at $t = 32$ (out of our model's maximum timestep of $T = 1000$). To ease compute requirements, we use batch size $B = 5$ for all experiments.

| | | SVHN | CelebA | GTSRB | CIFAR-10 | ImageNet32 |
|---|---|---|---|---|---|---|
| typicality ($t=8$) | SVHN | - | 0.9937 | 0.7122 | **0.9975** | **0.9969** |
| | CelebA | **1.0000** | - | 0.8816 | 0.3443 | **0.5904** |
| | GTSRB | **0.9990** | 0.8108 | - | 0.6680 | 0.7214 |
| | CIFAR-10 | **1.0000** | 0.8684 | **0.9172** | - | 0.4852 |
| | ImageNet32 | **1.0000** | 0.9869 | **0.9845** | 0.8800 | - |
| ours ($t=8$) | SVHN | - | **1.0000** | **0.9833** | 0.8214 | 0.9838 |
| | CelebA | 0.9932 | - | **0.9284** | **0.8250** | 0.3942 |
| | GTSRB | 0.9822 | **0.9998** | - | **0.7025** | **0.7485** |
| | CIFAR-10 | 0.9940 | **0.9998** | 0.8750 | - | **0.5071** |
| | ImageNet32 | 0.9934 | **1.0000** | 0.9624 | **0.9046** | - |
| typicality ($t=64$) | SVHN | - | 0.9813 | 0.4779 | **0.9970** | **0.9982** |
| | CelebA | **1.0000** | - | **0.9746** | 0.3288 | **0.6622** |
| | GTSRB | **0.9966** | 0.7805 | - | 0.6914 | **0.8543** |
| | CIFAR-10 | **1.0000** | 0.9228 | **0.9798** | - | **0.5388** |
| | ImageNet32 | **1.0000** | 0.9907 | **0.9965** | 0.8059 | - |
| ours ($t=64$) | SVHN | - | **1.0000** | **0.9801** | 0.9345 | 0.9603 |
| | CelebA | 0.9786 | - | 0.8856 | **0.6389** | 0.5063 |
| | GTSRB | 0.9762 | **0.9990** | - | 0.7842 | 0.7150 |
| | CIFAR-10 | 0.9825 | **0.9996** | 0.7911 | - | 0.5039 |
| | ImageNet32 | 0.9850 | **0.9999** | 0.9255 | 0.8001 | - |
| typicality ($t=512$) | SVHN | - | 0.7125 | 0.5117 | **0.9574** | **0.9839** |
| | CelebA | **0.9997** | - | **0.9984** | 0.3592 | 0.4543 |
| | GTSRB | 0.9549 | 0.7854 | - | 0.7207 | **0.8194** |
| | CIFAR-10 | **0.9997** | 0.9815 | **0.9971** | - | **0.5154** |
| | ImageNet32 | **0.9998** | 0.9908 | **0.9984** | 0.6338 | - |
| ours ($t=512$) | SVHN | - | **0.9962** | **0.9712** | 0.8089 | 0.7857 |
| | CelebA | 0.7654 | - | 0.9396 | **0.5205** | **0.5272** |
| | GTSRB | 0.6870 | **0.9496** | - | **0.7770** | 0.6256 |
| | CIFAR-10 | 0.6295 | **0.9844** | 0.9326 | - | 0.4660 |
| | ImageNet32 | 0.6635 | 0.9757 | 0.8904 | 0.5432 | - |

Table 3: auc values for typicality [top] and ours [bottom], batch size 5 applied to diffusion models for varied timesteps $t = 8, 64, 512$

### B.4.4 Ablation study on multiple q-samples for anomaly detection with diffusion models

In this section, we investigate if any performance improvement can be achieved by using multiple samples from $q$ to estimate $L_0$, ie $l(\boldsymbol{x}) = l(\boldsymbol{x}_0) = \mathbb{E}_{\boldsymbol{x}_1 \sim q(\boldsymbol{x}_1|\boldsymbol{x}_0)} \log p^{\boldsymbol{\theta}}(\boldsymbol{x}_0|\boldsymbol{x}_1) \ \mathbb{E}L_0 \propto \mathbf{E}_{\boldsymbol{x}_1 \sim q(\boldsymbol{x}_0)} \left\| \boldsymbol{\epsilon} - \boldsymbol{\epsilon}^{\boldsymbol{\theta}}(\boldsymbol{x}_1, 1) \right\|$. Specifically, we take $n = 5$ q-samples $\boldsymbol{\epsilon}^{(1)} \ldots \boldsymbol{\epsilon}^{(5)}, \boldsymbol{x_1^{(1)}} \ldots \boldsymbol{x_1^{(5)}}$ to define our likelihood proxy as:

$$l(\boldsymbol{x}) = \frac{1}{5} \sum_{i=1}^{5} \left\| \boldsymbol{\epsilon}^{(i)} - \boldsymbol{\epsilon}^{\boldsymbol{\theta}}(\boldsymbol{x_1^{(i)}}, 1) \right\|.$$

The AUROC values using batch size $B = 5$ for our method and typicality Nalisnick et al. (2019b) are in table 4. We note little to no performance gain for our method or typicality, motivating our use of $n = 1$ q-sample in our implementation for efficiency.

|  |  | SVHN | CelebA | GTSRB | CIFAR-10 | ImageNet32 |
|---|---|---|---|---|---|---|
| typicality | SVHN | - | 0.9251 | 0.9668 | 0.9824 | **0.9926** |
|  | CelebA | **0.9981** | - | 0.6328 | 0.5312 | 0.5500 |
|  | GTSRB | **0.9961** | 0.6768 | - | 0.6291 | 0.4716 |
|  | CIFAR-10 | **0.9685** | 0.7890 | 0.4130 | - | **0.7174** |
|  | ImageNet32 | **0.9959** | 0.5954 | 0.6098 | **0.7923** | - |
| ours | SVHN | - | **0.9984** | **0.9930** | **0.9873** | 0.9756 |
|  | CelebA | 0.8938 | - | **0.9746** | **0.8140** | **0.6952** |
|  | GTSRB | 0.8222 | **0.9823** | - | **0.9367** | **0.8728** |
|  | CIFAR-10 | 0.9683 | **0.9744** | **0.8922** | - | 0.5666 |
|  | ImageNet32 | 0.9797 | **0.9793** | **0.9188** | 0.7485 | - |

Table 4: auc values for typicality [top] and ours [bottom], batch size 5 applied to a diffusion model using 5 q-samples
average performance for typicality: 0.7617,    25/50/75 quantiles: 0.6062 / 0.7532 / 0.9720
average performance for ours: 0.8987,    25/50/75 quantiles: 0.8601 / 0.9525 / 0.9794

### B.5 Using Fisher's method in the place of density estimation

In this section, we briefly investigate the use of Fisher's method Fisher (1938) to compute the final anomaly score when using the gradient $L^2$-norm statistics $f^\ell$ which we define in §3.4. Specifically, if we modify our method by defining $q^\ell(\boldsymbol{x}) = \min(\Phi(f^\ell(\boldsymbol{x})), 1 - \Phi(f^\ell(\boldsymbol{x})))$ to be the $\ell$-th p-value from a two-tailed z-test and our final anomaly score as:

$$S = -\sum_{\ell=1}^{L} \log(q^\ell(\boldsymbol{x})).$$

In table 5 for brevity we only report our results for a Glow model with $B = 1$, but note the same pattern across all models. We note a small performance detriment across all dataset pairings from using Fisher's method, motivating our use of density estimation.

|  |  | SVHN | CelebA | GTSRB | CIFAR-10 | ImageNet32 |
|---|---|---|---|---|---|---|
| ours (Fisher) | SVHN | - | 0.9808 | 0.9494 | 0.8358 | 0.7514 |
|  | CelebA | 0.9633 | - | 0.8125 | 0.5022 | 0.2686 |
|  | GTSRB | 0.9321 | 0.9772 | - | 0.7016 | 0.4482 |
|  | CIFAR-10 | 0.9398 | 0.9250 | 0.8119 | - | 0.4203 |
|  | ImageNet32 | 0.9899 | 0.9734 | 0.9165 | 0.6969 | - |
| ours (density) | SVHN | - | **0.9880** | **0.9858** | **0.8747** | **0.8010** |
|  | CelebA | **0.9823** | - | **0.9262** | **0.5155** | **0.2997** |
|  | GTSRB | **0.9537** | **1.0000** | - | **0.7546** | **0.9967** |
|  | CIFAR-10 | **0.9658** | **0.9462** | **0.9126** | - | **0.4377** |
|  | ImageNet32 | **0.9976** | **0.9876** | **0.9683** | **0.7375** | - |

Table 5: auc values for ours (Fisher's method) [top] and ours (density estimation) [bottom], batch size 1 applied to glow
average performance for Fisher's method: 0.7898,    25/50/75 quantiles: 0.7004 / 0.8761 / 0.9529
average performance for density estimation: 0.8516,    25/50/75 quantiles: 0.7894 / 0.9500 / 0.9863

| Method | MNIST | Omniglot |
|---|---|---|
| WAIC | 0.766 | 0.796 |
| S using PixelCNN++ and FLIF | 0.967 | 1.000 |
| PixelCNN Gradient norms (OneClassSVM) (ours) | 0.979 | 1.000 |
| S using Glow and FLIF | 0.998 | 1.000 |
| Glow Gradient norms (OneClassSVM) (ours) | 0.819 | 1.000 |

Table 6: Table of results comparing the performance of our method on for a model trained on FashionMNIST at detecting OOD grayscale images to the performances of the S-score reported in Serrà et al. (2020) and Watanabe-Akaike Information Criterion reported in Choi et al. (2018)

## C  Supervised gradient-based methodology for classifiers

For completeness, we discuss classifier based OOD detection methods using the gradient, noting that these methods *are* given label-information at train time and our representation-invariance result does not directly translate over to this paradigm.

In order to compute gradients without a target label, This approach is hence supervised with respect to in-distribution and OOD labels which it requires. Liang et al. (2018) propose a method called ODIN which uses the gradient with respect to the *data*: They backpropagate gradients to the input data to see how much an input perturbation can change the softmax output of a classifier, following the intuition that OOD inputs may be more sensitive and prone to a larger variation in the output distribution. Igoe et al. (2022) are critical of the use of classifier gradients, instead advocating that most information can be recovered from the layer representations. Behpour et al. (2023) propose projection of the gradient onto the space generated by in-distribution gradients, motivated as in Kwon et al. (2020) by the informativity of the gradient angle for OOD detection.

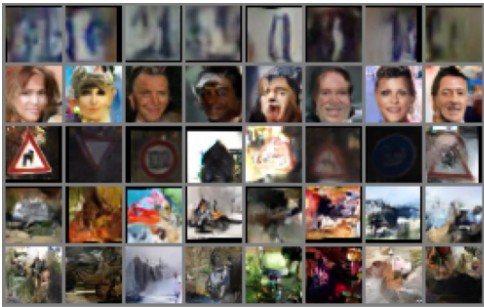

Figure 18: *Samples from Glow models.* Samples from the Glow models used in our Experiments, trained on `SVHN`, `CelebA`, `GTSRB`, `CIFAR-10` and `ImageNet32` respectively from top to bottom.

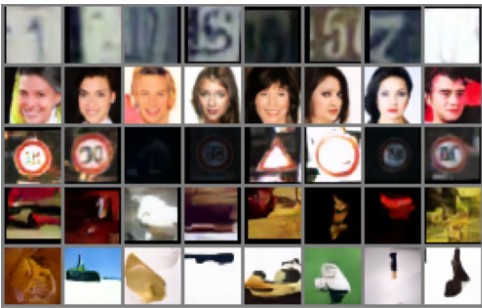

Figure 19: *Samples from diffusion models.* Samples from the denoising diffusion models used in our Experiments, trained on `SVHN`, `CelebA`, `GTSRB`, `CIFAR-10` and `ImageNet32` respectively from top to bottom.

## D   Code, models

Our *Glow* implementation derives from a repository `https://github.com/y0ast/Glow-PyTorch` replicating the one used in Nalisnick et al. (2019a), with the only difference being we use a batch size of 64 in training rather than 512. See Figure 18 for samples from our models.

Our diffusion model implementation derives from a PyTorch transcription at `https://github.com/lucidrains/denoising-diffusion-pytorch` of that described in Ho et al. (2020). We train using Adam with a learning rate of $3e^{-4}$ for 10 epochs. Our model has $T = 1000$ timesteps which are uniformly sampled from in training and the U-Net backbone has dimension multiplicities of $(1, 2, 4, 8)$. See Fig 19 for samples from our models.

Our code is available with pre-trained models and pre-computed gradient $L^2$ norms at `https://github.com/SamD770/Generative-Models-Knowledge`.

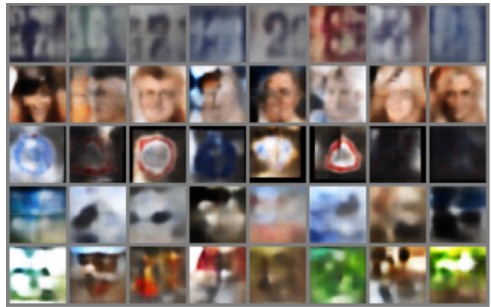

Figure 20: *Samples from VAE models.* Samples from the denoising diffusion models used in our Experiments, trained on `SVHN`, `CelebA`, `GTSRB`, `CIFAR-10` and `ImageNet32` respectively from top to bottom.

## E   Results for poorly performant VAE

| test ↓ train → | | SVHN | CelebA | GTSRB | CIFAR-10 | ImageNet32 |
|---|---|---|---|---|---|---|
| typicality (B = 1) | SVHN | - | 0.5680 | 0.4302 | 0.5664 | **0.5933** |
| | CelebA | **0.4569** | - | 0.3729 | **0.5085** | **0.4868** |
| | GTSRB | 0.5901 | 0.6290 | - | 0.6484 | 0.6061 |
| | CIFAR-10 | 0.4223 | 0.4742 | 0.3454 | - | **0.4922** |
| | ImageNet32 | 0.4429 | 0.4831 | 0.3736 | 0.5047 | - |
| ours (B = 1) | SVHN | - | **0.6693** | **0.5541** | **0.5904** | 0.5740 |
| | CelebA | 0.4329 | - | **0.4337** | 0.4685 | 0.4588 |
| | GTSRB | **0.5920** | **0.6581** | - | **0.6570** | **0.6629** |
| | CIFAR-10 | **0.4343** | **0.5826** | **0.5123** | - | 0.4864 |
| | ImageNet32 | **0.4582** | **0.5941** | **0.5048** | **0.5187** | - |
| typicality (B = 5) | SVHN | - | 0.9978 | 0.7943 | **0.9975** | **0.9961** |
| | CelebA | **1.0000** | - | 0.7642 | 0.3156 | 0.3621 |
| | GTSRB | **0.9998** | 0.8336 | - | 0.6809 | 0.5765 |
| | CIFAR-10 | **1.0000** | 0.7808 | **0.8332** | - | 0.4488 |
| | ImageNet32 | **1.0000** | 0.9866 | **0.9675** | **0.9266** | - |
| ours (B = 5) | SVHN | - | **1.0000** | **0.9970** | 0.8457 | 0.9561 |
| | CelebA | 0.9908 | - | **0.8552** | **0.7734** | **0.4202** |
| | GTSRB | 0.9716 | **0.9997** | - | **0.7325** | **0.9007** |
| | CIFAR-10 | 0.9895 | **0.9992** | 0.8104 | - | **0.5733** |
| | ImageNet32 | 0.9862 | **1.0000** | 0.9309 | 0.8532 | - |

Table 7: *VAE models* Comparison of the AUROC values (larger values are better) of our method to the typicality test Nalisnick et al. (2019b) for batch sizes $B = 1, 5$. We train VAE Kingma & Welling (2014) models on five natural image datasets (as columns) and evaluate the ability of the model-method combination to reject the other datasets (as rows).

Our VAE Kingma & Welling (2014) implementation uses entirely convolutional layers, in Figure 20 we note that the samples produced approximate the colour palate of the train datasets well but have poor semantic coherence. In table 7 we note poor performance for both our methods using this models as a backbone.

