# OpenReview forum: "Approximations to the Fisher Information Metric of Deep Generative Models for Out-Of-Distribution Detection"
_TMLR — Accepted by TMLR_

### Review · Reviewer_BFSA · 2024-04-29

**Summary Of Contributions:**

The paper discusses the problem of out of distribution detection by generative models. The paper proposes using the gradient of the model parameters w.r.t. data to distinguish out of distribution samples. It argues that the gradient at a local minima will be small for in distribution samples and high for out of distribution ones.

The paper also provides an empirically efficient algorithm to approximate said gradient for out of distribution detection.

**Audience:**

Yes

**Claims And Evidence:**

Yes

**Requested Changes:**

In addition to addressing the weaknesses enumerated above, the discussion under "likelihood ratio" subsection of section 2.2 is not at all clear. I'm not sure what point is being made here and it is not clear why likelihood ratio would be a suboptimal way for OOD detection.

**Strengths And Weaknesses:**

Strengths:
- The paper provides a principled approach to out of distribution detection by generative models. It proposes to use size of gradient information to detect such samples.
- Further, it provides an empirically efficient algorithm to approximate such high dimensional gradients via approximation of the Fisher information matrix by diagonally dominant matrices. This essentially boils down to looking at the l2 norms of the layer wise gradients.

Weaknesses:
- Providing a more principled definition of what OOD data would satisfy the assumptions of the paper would be useful. It is not clear that the gradient would be large for any OOD sample.
- It is not theoretically clear to me why likelihood estimation is a poor method for OOD detection. Perhaps adding a short explanation/mathematical proof will provide the paper more grounding in the literature.

---

> ### Author Response · Authors · 2024-06-28
> **OOD data should be "unlearned"**
>
> > "Providing a more principled definition of what OOD data would satisfy the assumptions of the paper would be useful. It is not clear that the gradient would be large for any OOD sample"
>
> Thank you for asking this clarifying question. We expect for the size of the $L^2$ norm of the gradient at layer l to be proportional to the gain in likelihood from a very small SGD update. Colloquially, our definition of novel/OOD dictates that samples which could have their likelihoods greatly improved by a small gradient update are OOD. In the revision, we have included this explanation in $\S 3.1$.
>
> Empirically, defining an OOD detection method is equivalent to defining the set of OOD data. The quality of this definition depends on if it lines up with the semantic intuition of what OOD means, which we have validated for our method using our evaluation criterion.

---

> ### Author Response · Authors · 2024-06-28
> **Likelihood estimation fails empirically, there are some theoretical results prior works use to convey intuition.**
>
> > "It is not theoretically clear to me why likelihood estimation is a poor method for OOD detection. Perhaps adding a short explanation/mathematical proof will provide the paper more grounding in the literature."
>
> Ultimately the poor performance of likelihood estimation is based on empirical observations of model behavior, rather than theory.
>
> Empirically, Nalisnick et al. (2019a) and Choi et al. (2018) concurrently demonstrated that normalizing flows an VAEs trained on CIFAR-10 will assign higher likelihood values to SVHN, this is a result we replicate in Figure 2 and extend to diffusion models. Following these results, Schirrmeister et al. (2020) and Serrà et al. (2020) demonstrated that this phenomenon can occur because generative models tend to assign higher likelihoods to data points with less “complexity” a priori, and thus low-complexity out-of-distribution data can override more complex in-distribution data. Perhaps this can theoretically be intuited by how, for discrete data, the model’s negative log likelihood is directly proportional to the number of bits needed to encode the data under arithmetic coding.
>
> One theoretical result that Nalisnick et al. (2019b) reference is the gaussian annulus result Blum et al. (2020), the essence of which can be observed by considering strings of Bernoulli variables. Consider a dataset of binary strings $x_1, x_2 \dots x_N \in \mathbb{R}^{10,000}$, where each component from the string is and independently and identically distributed bernoulli random variable $x_{nd} \sim^{iid} \text{Bernoulli}(0.6)$. Define the sum of a binary string to be the number of ones it contains:
>
> $$S(x_n) = \\sum_{i=d}^D x_{nd}$$.
>
> We see that S(x_d) will be binomially distributed with mean $6000$ and standard deviation $10 \sqrt{24} \approx 50$. We thus see that the vast majority of the in-distribution data will have a sum of $6000 \pm 150$, even though the most likely binary string has a sum of 10,000 (as it contains all 1 values).
>
> In  $\S 2.2$ of the revision, we expanded our discussion of the gaussian annulus result and the relation of the likelihood to arithmetic coding to give these empirical observations a more detailed theoretical background.

---

> ### Author Response · Authors · 2024-06-28
> **Likelihood ratios require a model of the OOD distribution.**
>
> > In addition to addressing the weaknesses enumerated above, the discussion under "likelihood ratio" subsection of section 2.2 is not at all clear. I'm not sure what point is being made here and it is not clear why likelihood ratio would be a suboptimal way for OOD detection."
>
> Likelihood ratios require a certain amount of access to a pre-specified OOD distribution, limiting their applicability more than our method or typicality. In the revision, we have revised $\S 2.2$ to increase the clarity of this point.
>
> The difficulty with using likelihood ratios is that they require choosing an OOD distribution $p^{OOD}$ and access to some function $\mathbf{x} \mapsto \log p^{OOD}(\mathbf{x})$ that evaluates the likelihood of a datapoint under this distribution. The set of applications where we have access to such a function but not to an explicit sample of OOD data (at which point a classification approach becomes more appropriate) is limited.

---

### Review · Reviewer_SnJ5 · 2024-05-17

**Summary Of Contributions:**

This paper proposes a method for OOD detection using likelihood-based deep generative models. Specifically, the method uses the layer-wise gradient norms as features, fits a Gaussian distribution, and uses the negative log likelihood of the fitted Gaussian as the OOD score. Even though likelihood-based deep generative models are designed to model the data distribution, prior work finds that they assign higher likelihood to OOD data. The proposed method builds on the intuition that if the model is fitted on some distribution of data, then seeing any OOD data should affect the model and induce a large gradient.

The method is theoretically motivated by the Fisher information metric, and can be seen as an efficient approximation of it. Authors conduct experiments for both the Glow model and the Diffusion model, and compare their method to the standard typicality test on a number of datasets.

**Audience:**

Yes

**Broader Impact Concerns:**

None.

**Claims And Evidence:**

Yes

**Requested Changes:**

The following suggestions are not critical to my recommendation, but would strengthen the paper
- Providing more robust justification for using the layer-wise gradient norm as a signal (see weaknesses)
- Providing more justification for using one inference step on the diffusion model. How does this affect the performance of the method?
- See question about hat-value.

**Strengths And Weaknesses:**

Strengths
- The paper makes a contribution of evaluating gradient-based methods for OOD detection on deep generative models, and connecting their approach to the Fisher Information Metric.
- The experiments are comprehensive.
- The paper is clearly written and easy to follow.

Weaknesses
- The method is model-agnostic, but it doesn’t work as well on diffusion models compared to the Glow model: for the diffusion model experiment, the proposed method is better only around half the time. In fact, this can be seen in the experiments in Appendix B.2.2 for the gtsrb, cifar10, and imagenet3 results: the intuition that the score is highly informative for OOD detection is not validated in these plots.
- For diffusion models the one-step log-likelihood is used, which is very different from the actual log-likelihood. It would be helpful to have some exposition on how this affects the performance of the method, or whether there are other approaches to apply the proposed method to diffusion models.
- Regarding novelty, as authors suggested, similar methods have been independently discovered in other works and other contexts.

Question: what’s the relationship between the hat-value and influence function? Are they the same?

---

> ### Author Response · Authors · 2024-06-28
> **Randomly selected layers may not be informative for OOD detection**
>
> > “The method is model-agnostic, but it doesn’t work as well on diffusion models compared to the Glow model: for the diffusion model experiment, the proposed method is better only around half the time.“
>
> > “In fact, this can be seen in the experiments in Appendix B.2.2 for the gtsrb, cifar10, and imagenet3 results: the intuition that the score is highly informative for OOD detection is not validated in these plots.”
>
> We kindly disagree with the conclusions drawn from Appendix B. 2.2 and would like to clarify this potential misunderstanding:   In Appendix B. 2.2, we show the layer-wise score for ***randomly selected layers***. We wouldn’t expect all of them to be informative for OOD detection. It is perfectly sufficient for a subset of them to be highly informative for performing OOD detection well, for instance with our method or say a classifier trained on layer-wise scores as features.
>
> Furthermore, as we mention in $\S 5$, we shouldn’t necessarily expect models trained on semantically diverse datasets like CIFAR-10 and ImageNet32 (the last two columns in tables 1 & 2) to be able to reject others (for example, a cat image from CIFAR-10 may well lie within the distribution of cats in ImageNet32), and mainly include these benchmarks for consistency with prior work. However, we agree that this is an important discussion that could be extended. In our revised manuscript, we reworded our criticism of using these datasets as training distributions in benchmarks and explicitly state that consistency is the reason for including them. Examining the results for models trained on SVHN, CelebA and GTSRB (the first three columns in tables 1 & 2), we see that the diffusion models trained on CelebA and GTSRB generally outperform typicality, at the cost of poorer performance on SVHN.

---

> ### Author Response · Authors · 2024-06-28
> **Our ablation experiments indicate that the one-step log-likelihood provides the best performance of the likelihood proxies tried.**
>
> >  “For diffusion models the one-step log-likelihood is used, which is very different from the actual log-likelihood. It would be helpful to have some exposition on how this affects the performance of the method, or whether there are other approaches to apply the proposed method to diffusion models.”
>
> For diffusion models, using full variational lower bound on the log-likelihood is prohibitively expensive to use, so we elect to use one of its components. In appendix B.4. we perform an ablation study on which component is most effective for both our method and typicality, finding the one-step likelihood to be most efficacious, and empirically observe these components behave similarly to the full likelihood with respect to the phenomenon observed by Nalisnick et al. (2019a). We do agree that there is a wider set of likelihood proxies for diffusion models that could be experimented with, but defer this for future work.
>
> As we write in equation (8) of appendix B.4.1, there is an evidence lower bound on the likelihood derived by Ho et al, given by:
>
> $$\log p^{\theta}(x) >  \sum_{t=0}^T \mathbb{E} \left(L_t \right)$$
> We would like to note that even doing a one-sample Monte-Carlo approximation of the right hand side of this equation would require $T=1000$ forward-backward passes per sample, meaning that doing one evaluation on one dataset would have, at a minimum, an equivalent cost to 1000 epochs of training (in our implementation this is multiplied by the training batch size (32x) greater as there is no native PyTorch implementation for computing batched per-sample gradients). Running such an eval is outside our compute budget, and would likely be outside the compute/latency budget of many applications, so we looked at using individual samples $L_t$ .
>
> In Appendix B.4.3, we perform an ablation of using differing values of t (with higher t corresponding to adding more noise) on the performance of our method, noting that the lower values of t give superior performance. In Appendix B.4.4, we find no performance improvement to using a multi-sample Monte Carlo approximation of $\mathbb{E}(L_0)$. These empirical performance measurements are ultimately the reason for the choice of $L_0$.
>
> In Figures 2 & 16 we observe these $L_t$ values to behave similarly to the likelihood, assigning high likelihoods to certain datasets in the same pattern as observed in Nalisnick et al. (2019a).

---

> ### Author Response · Authors · 2024-06-28
> **Novelty**
>
> > "Regarding novelty, as authors suggested, similar methods have been independently discovered in other works and other contexts."
>
> We agree that the resultant method is not very different from prior work, and that this is a weakness of the paper. Nonetheless, we believe our theoretical justifications ($\S 2.2$), empirical investigations into the approximations made and their validity ($\S 3$) and our extensive evaluation including diffusion models ($\S 5$), will be of interest to the research community.
>
> As we also state in our response to reviewer U1Sv, a list of contributions of the paper is:
> - In $\S 2.2$: Theoretically demonstrating that the score vector satisfies the principle of invariance proposed by Le Lan & Dinh (2021).
> In Figures 2 [Right] and 16: Empirically demonstrating that the t-step likelihoods of denoising diffusion models exhibit the phenomenon demonstrated by Nalisnick et al. (2019a).
> - In $\S 3.3$ : Empirically demonstrating that the FIM of Generative Flows and Diffusion models are sufficiently well behaved intra-layer such that the layer-wise fisher information metric admits a very efficient approximation (simply taking the $L^2$-norm).
> - In $\S 3.4$ and Figure 4: Empirically demonstrating that the log gradient norms take the normal distribution that the theory predicts, in spite of the approximations made (the theory assumes a perfect model distribution, a complete FIM, etc.).
> - In $\S 3.5$: Leveraging this theoretically predicted and empirically verified normal distribution in our method.
> - In $\S 4:$ Linking prior works that have independently found the gradient to be informative for OOD detection.
> - In $\S 5:$ Our evaluation is more extensive than prior works. Kwon et al. (2020), Choi et al. (2021), Bergamin et al. (2022) all only evaluate color datasets using VAEs or normalizing flows trained on CIFAR-10. Providing experimental results using 5 different training distributions gives a measure of performance which is much more robust to specificities of the training/evaluation datasets.
> Providing guidance for applying the method to diffusion models, as well as robust performance measures, takes the method closer to practical uses of generative models.

---

> ### Author Response · Authors · 2024-06-28
> **Relationship between the hat-value and influence function**
>
> > "Question: what’s the relationship between the hat-value and influence function? Are they the same?"
>
> Hat values are specific to linear regression, in contrast influence functions are a technique that can be applied to any statistical model and estimator. In the below, we derive a relation between them.
> Assume the standard linear regression setup, with a design matrix X and target vector Y comprised of observations $(x_i, y_i)$, which learns the parameter vector $\\hat{\\beta} = (X^T X)^{-1} X Y$
> The hat value is defined as the ith diagonal element of the hat matrix:
>
> $$h_{ii} = (X^T (X^T X)^{-1}X)_{ii} = x_i^T(X^T X)^{-1}x_i$$.
>
> This can be shown to equal the derivative of the ith prediction with respect to the ith target:
>
> $$h_{ii} = \\frac{\hat{y}_i}{y_i}$$
>
> Influence functions measure the marginal influence of introducing a new datapoint to the training dataset. Consider adding a new point $(x, y)$ with some small weighting t to a linear regression dataset $(x_i, y_i)$, which we reweight to a factor of (1 - t), the new maximum likelihood estimate for the parameter vector $\hat{\beta}_t(x, y)$ satisfies:
>
> $$\left((1 - t)X^T X + t x^t x \right) \hat{\beta}_t(x, y) = (1 - t)X^T Y$ + t x^T y.$$
>
> We can compute the influence function denoted $IF_{\hat{\beta}}(x, y)$, to be:
>
> $$IF_{\hat{\beta}}(x, y) := \left. \frac{\partial \hat{\beta}_t(x, y)}{\partial t} \right|_0 = (X^T X)^{-1}x^T(y - x \hat{\beta})$$
>
> We can thus see that the concepts are linked by the fact that the influence function on the linear model’s ith prediction, denoted y_i^, evaluated at the true ith datapoint (x_i, y_i), will be equal to the product of the ith hat value, h_{ii}, and the ith residual, $e_i = y_i - x_i \\hat{\\beta}$ ie:
>
> $IF_{\\hat{y}_i}(x_i, y_i) $ =
>
> $x_i^T IF_{\\hat{\\beta}}(x_i, y_i) $ =
>
> $ x_i^T (X^T X)^{-1}x_i^T e_i = h_{ii}e_i$
>
> where we have used the chain rule for Gateaux derivatives.

---

### Review · Reviewer_U1Sv · 2024-06-14

**Summary Of Contributions:**

The authors propose a way to perform out-of-distribution (OOD) detection using a pre-trained generative model $p_\theta$. To assess if a point $x$ is OOD, the key idea is to use the magnitude of the gradient of the log-likelihood $\nabla_\theta \log p_\theta(x)$. Instead of using the standard L2 norm, they argue (using both heuristics and a theoretical argument of invariance) that this magnitude should be measured using the Mahalanobis distance induced by the Fisher information matrix (FIM) of $p_\theta$. They discuss ways of approximating the FIM, and use their score to perform OOD detection for diverse kinds of generative models: normalizing flows, diffusion models, variational auto encoders, autoregressive models.

**Audience:**

Yes

**Broader Impact Concerns:**

I have no particular concern.

**Claims And Evidence:**

Yes

**Requested Changes:**

The authors should clarify what additional scientific insight their paper provide compared to Choi et al. (2021) and Bergamin et al. (2022). One of these differences is the invariance property (Proposition 1). They should clarify how it is related to classical invariance properties of likelihood ratios and scores (e.g. De Maio, 2000, and references herein). The authors should mention earlier in the manuscript that these earlier works used the FIM in the same way, albeit with different motivations (here, the first mention of  is on page 11). What new things do the experiments of the authors bring? How are their experiments different from prior work? If the FIM approximation is indeed the only important difference, maybe doing experiments on its influence could be interesting.


# Additional reference

De Maio et al., On the Invariance, Coincidence, and Statistical Equivalence of the GLRT, Rao Test, and Wald Test, IEEE Trans. on Signal Processing, 2010

**Strengths And Weaknesses:**

# Strengths

The idea of using the FIM to reweigh the individual gradients of the test points is very well motivated, both by compelling heuristics and by an invariance argument (Proposition 1). The authors perform lots of illustrative experiments that are sometimes insightful (e.g. Figs 5 and 6). To the best of my knowledge, they are also the first to conduct these kinds of experiments for diffusion models.

# Weaknesses

My main concern with this paper is that the idea of using gradients weighted by the FIM, as well as most of the insights provided in the paper, seemed to be already put forward by Choi et al. (2021) and Bergamin et al. (2022), as acknowledged by the authors in Section 4. Choi et al. (2021) used a Bayesian motivation, and Bergamin et al. (2022) the point of view of statistical tests, but their final OOD scores are virtually identical to the one proposed in this paper (the only important difference being the way to approximate the FIM). Clarifying what the paper brings additionally to the table, compared to this prior works, would make it more useful. As I mention in the "requested changes" section, the invariance property should also be contextualized with respect to the literature.


# Less important issues and unclear things

- Introduction (and elsewhere) : the authors mention that the fact that models give higher likelihoods to OOD “marks an open problem for deep generative models“. However, as they briefly discuss in Section 2.2, the problem is not limited to deep generative models, and a simple (high-dimensional) Gaussian behaves exactly the same way.

- Introduction : "In §2.2) we will analyse the small amount of previous work on gradient-based OOD detection, linking seemingly independent discoveries of other authors." What links drawn here are novel? And why not cite Choi et al. (2021) and Bergamin et al. (2022) in this §2.2)? They are surely among the "small amount of previous work on gradient-based OOD detection".

- Section 3.5 : The authors claim that diffusion models “do not allow for exact inference of the log-likelihood”. This is not entirely true, see for instance the ODE approach of Song et al. (2021)

- Section 4 : “Approximating the FIM as a diagonal, with coefficients $F_{\alpha \alpha} = (\delta_\alpha \log p_\theta(x) )^2 + \varepsilon$  . (…) We also note that layers whose gradient values are << 10−8 (see Appendix B.3 Figures 13 and 14) would have their information nullified without careful tuning of ε” Since 10-8 is added to the square of the gradient, shouldn’t that be 10^-4 ? Is that really source of concern? It would be interesting to do experiments with different approximations of the FIM.

- Section 4: "Nalisnick et al. (2019b) note that the Maximum Mean and Kernelized Stein Discrepancy tests they use to benchmark their typicality test are only effective when using the Fisher kernel Jaakkola & Haussler (1998); Jacot et al. (2018) $k(xi,xj) = \nabla_\theta l (x_i) \nabla_\theta l (x_j)^T$". Nalisnick et al. (2019b) used a modified version of the Fisher kernel, that was actually $ \nabla_x l (x_i) \nabla_x l (x_j)^T$.

- Section 4: "Our method differs from using Fisher kernel in that it allows for information to be used from all the layers, rather than a few dominant layers" With the standard Fisher kernel from Jaakkola & Haussler (1998), the FIM also appears, and makes sure that all weights are properly normalized. Moreover, Bergamin et al. (2022) showed that MMD with the Fisher kernel is essentially equivalent to using your criterion.

- Appendix A.3: The authors could explain the Explain better the “snake pattern”. The formula is a bit cryptic. Plotting digits would be hepful.

- Proposition 2: “then, ELBO is invariant”, shouldn’t it be “the gradient of the ELBO is invariant”?

- Proposition 3: There is a “< alpha” missing, both in the equation of the Prop, and in the proof

# Typos


Beginning of Section 2.2: “negative leaned likelihood”

Title of Appendix 1.2: “variatonal”


# Additional reference

Song et al., Score-Based Generative Modeling through Stochastic Differential Equations, ICLR 2021

---

> ### Author Response · Authors · 2024-06-28
> **Novelty**
>
> > “My main concern with this paper is that the idea of using gradients weighted by the FIM, as well as most of the insights provided in the paper, seemed to be already put forward by Choi et al. (2021) and Bergamin et al. (2022), as acknowledged by the authors in Section 4. Choi et al. (2021) used a Bayesian motivation, and Bergamin et al. (2022) the point of view of statistical tests, but their final OOD scores are virtually identical to the one proposed in this paper (the only important difference being the way to approximate the FIM). Clarifying what the paper brings additionally to the table, compared to this prior works, would make it more useful. [...]”
>
> > “What new things do the experiments of the authors bring? How are their experiments different from prior work? If the FIM approximation is indeed the only important difference, maybe doing experiments on its influence could be interesting.”
>
> We agree that at this point novelty is the main weakness of the paper, however we believe the contributions are sufficient to be of interest to the wider community.
>
> As we also state in our response to reviewer SnJ5, a list of contributions of the paper is:
> - In $\S 2.2$: Theoretically demonstrating that the score vector satisfies the principle of invariance proposed by Le Lan & Dinh (2021), see “comparison to classical group-invariance” below for how the theoretical setup used by Le Lan & Dinh (2021) is incompatible with that used in eg. DeMiao et al. (2010).
> In Figures 2 [Right] and 16: Empirically demonstrating that the t-step likelihoods of denoising diffusion models exhibit the phenomenon demonstrated by Nalisnick et al. (2019a).
> - In $\S 3.3$: Empirically demonstrating that the FIM of Generative Flows and Diffusion models are sufficiently well behaved intra-layer such that the layer-wise fisher information metric admits a very efficient approximation (simply taking the $L^2$-norm).
> - In $\S 3.4$ and Figure 4: Empirically demonstrating that the log gradient norms take the normal distribution that the theory predicts, in spite of the approximations made (the theory assumes a perfect model distribution, a complete FIM, etc.).
> - In $\S 3.5$: Leveraging this theoretically predicted and empirically verified normal distribution in our method.
> - In $\S 4$: See “linking prior works” below.
> - In $\S 5$: Our evaluation is more extensive than prior works. Kwon et al. (2020), Choi et al. (2021), Bergamin et al. (2022) all only evaluate color datasets using VAEs or normalizing flows trained on CIFAR-10. Providing experimental results using 5 different training distributions gives a measure of performance which is much more robust to specificities of the training/evaluation datasets.
> Providing guidance for applying the method to diffusion models, as well as robust performance measures, takes the method closer to practical uses of generative models.

---

> ### Author Response · Authors · 2024-06-28
> **Comparison to classical group-based invariance**
>
> >”As I mention in the "requested changes" section, the invariance property should also be contextualized with respect to the literature.”
>
> >“They should clarify how it is related to classical invariance properties of likelihood ratios and scores (e.g. De Maio, 2000 [sic], and references herein).”
>
> The theoretical setup used by De Maio el al (2010), initially proposed by Lehmann (1986), substantively differs from that used in LeLan & Dinh (2021), which we use and argue is more applicable in the case of deep generative models.
>
> The setup proposed by Lehmann (1987) is one in which we consider a group of transformations from the  input space to itself $g: X \to X$ which are sufficiently narrow such that there is a corresponding group of transformations to the parameter space $\bar{g}: \Theta \to \Theta$ that counteract the effect of each transformation, formally defined by:
>
> $$\mathbb{P}^{\bar{g}\theta}_X = \mathbb{P}^{\theta}_X \circ g^{-1}.$$
>
> One example of where this setup is applicable is applying dilations to an input space of a multivariate normal distribution, whereby any linear dilation of the input space can be counteracted by a dilation of the covariance matrix. This is not the case for the arbitrary transformations $f$ considered in Proposition 1 of Le Lan & Dinh (2021), which we cannot guarantee to be counteracted by some transformation of a generative model’s parameters. Even the simple example of the non-linear RGB-HSV transformation we give in \S 2.3 can only approximately be counteracted by changing the generative model’s parameters.
> In contrast, the setup proposed by Le Lan & Dinh (2021), implicitly considers transformations from arbitrary measure spaces $f: X \to f(X)$, and considers the pullback:
>
> $$\mathbb{P}_{f(X)}^{\theta} = \mathbb{P}_X^{\theta} \circ f^{-1}$$
>
> The presence of the counteracting parameter transformation $\bar{g}$ in equation (1) ultimately leads to a presence of a $\frac{\partial \bar{g}^{-1}}{\partial \theta}$ term in the score vector (equation (6) of DeMaio et al. 2010), which does not appear when using the setup of Le Lan & Dinh (2021). This leads to substantive differences in the theory, notably that under the classical setup the score vector itself is not invariant.  In Appendix A.5 of the revision, we have included an expanded version of this discussion.
>
> Additional reference:
>
> E. L. Lehmann, Testing Statistical Hypotheses, ser. Springer Texts in Statistics, 2nd ed. New York: Springer-Verlang, 1986.

---

> ### Author Response · Authors · 2024-06-28
> **Linking prior works**
>
> > “Introduction : "In §2.2) we will analyse the small amount of previous work on gradient-based OOD detection, linking seemingly independent discoveries of other authors." What links drawn here are novel? And why not cite Choi et al. (2021) and Bergamin et al. (2022) in this §2.2)? They are surely among the "small amount of previous work on gradient-based OOD detection".”
>
> > “...The authors should mention earlier in the manuscript that these earlier works used the FIM in the same way, albeit with different motivations (here, the first mention of [it] is on page 11).”
>
> In the revision, we have amended a here typo directing to $\S 2.2$ instead of $\S 4$, thank you for noting this and apologies for the confusion caused. We have also cited these works in the introduction and in $\S 3.2$.
>
> Regarding the novelty of the links drawn; $\S 4$ we note five works which all, at least in part, perform experiments finding informativity using the gradients of the parameters of deep generative models for OOD detection, namely Nguyen et al. (2019), Kwon et al. (2020), Choi et al. (2021), Bergamin et al. (2022) and Appendix D of Nalisnick et al. (2019b) (see “Use of gradients in Nalisnick et al. (2019b)” below). Only one of these works cites one other’s experiments (namely Bergamin et al. (2022)’s citation of Choi et al. (2021)), so linking these independent findings provides a more robust case for the informativity of gradients.

---

> ### Author Response · Authors · 2024-06-28
> **Use of gradients in Nalisnick et al. (2019b)**
>
> > “Section 4: "Nalisnick et al. (2019b) note that the Maximum Mean and Kernelized Stein Discrepancy tests they use to benchmark their typicality test are only effective when using the Fisher kernel Jaakkola & Haussler (1998); Jacot et al. (2018) $k(xi,xj) = \nabla_\theta l (x_i) \nabla_\theta l (x_j)^T$". Nalisnick et al. (2019b) used a modified version of the Fisher kernel, that was actually $ \nabla_x l (x_i) \nabla_x l (x_j)^T$.”
>
> Thank you for pointing this out,:in the revision we have adapted this sentence to include what we interpret to be the meaning of the following statements in Nalisnick et al. (2019b).
>
> In Nalisnick et al. (2019b) Appendix D the authors state:
>
>  “We found that MMD and KSD only had good performance when using the Fisher kernel (Jaakkola & Haussler, 1999):  $k(x_i, x_j) = \nabla_{\theta}p(x_i; \theta)^T \nabla_{\theta}p(x_i; \theta)$  […] Hence in the experiments we use the kernel modified such that the derivative is taken w.r.t. the input (making it the likelihood score):  $k’(x_i, x_j) = \nabla_{x_i}p(x_i; \theta)^T \nabla_{x_j}p(x_i; \theta)$."
>
> Our reading of this statement is that they achieved good performance with the gradient with respect to the parameters but ultimately chose to use the gradient with respect to the data for lower memory costs (we presume they were caching the computed gradients for easier experimental iteration).

---

> ### Author Response · Authors · 2024-06-28
> **Nullification of small gradient values**
>
> > “Section 4 : “Approximating the FIM as a diagonal, with coefficients $F_{\alpha \alpha} = (\delta_\alpha \log p_\theta(x) )^2 + \varepsilon$ . (…) We also note that layers whose gradient values are << 10−8 (see Appendix B.3 Figures 13 and 14 [Figures 14 and 15 in the revision]) would have their information nullified without careful tuning of ε” Since 10-8 is added to the square of the gradient, shouldn’t that be 10^-4 ? Is that really source of concern? It would be interesting to do experiments with different approximations of the FIM.”
>
> Thank you for pointing out that we use the term “gradient values”, when we are in fact referring to plots of the FIM which are using the square of the gradient/product of two gradient values.  In the revision we have amended this.
>
> Perhaps a better illustration of the importance of these low-magnitude gradient signals is in Appendix B.2.1 figure 10. In this figure, we see that the diffusion model has many informative layers for which the aggregate $L^2$ norm of the entire layer takes values less than $< e^{-20} < 10^{-8}$, and thus individual squared gradient values will be less than 10^(-8).   In the revision we have referenced this figure too.

---

> ### Author Response · Authors · 2024-06-28
> **We disagree that the Gaussian annulus result completely explains the phenomenon of high OOD likelihoods.**
>
> > “Introduction (and elsewhere) : the authors mention that the fact that models give higher likelihoods to OOD “marks an open problem for deep generative models“. However, as they briefly discuss in Section 2.2, the problem is not limited to deep generative models, and a simple (high-dimensional) Gaussian behaves exactly the same way.”
>
> We respectfully disagree that the entire phenomenon of Nalisnick et al. (2019a) is explained by the gaussian annulus result, although this is hard to refute without being more specific. For example, the idea that there is some band of model likelihood that separates the semantically in-distribution samples from OOD sample isn’t validated by figure 2, where we observe many OOD samples with the same likelihood values as in-distribution samples.
>
> The problem of single-sample novelty recognition even using simple, semantically distinct train datasets such as GTSRB and CelebA, is still open.

---

> ### Author Response · Authors · 2024-06-28
> **Correction on exact likelihood inference for diffusion models.**
>
> > “Section 3.5 : The authors claim that diffusion models “do not allow for exact inference of the log-likelihood”. This is not entirely true, see for instance the ODE approach of Song et al. (2021)”
>
> Thank you for noting this, we have amended this statement in the revision.

---

> ### Author Response · Authors · 2024-06-28
> **Plot of the "snake pattern"**
>
> > “Appendix A.3: The authors could explain the Explain better the “snake pattern”. The formula is a bit cryptic. Plotting digits would be hepful. [sic]”
>
> In the revision, we have added a plot to Appendix A.3 illustrating this unraveling formula.

---

> ### Author Response · Authors · 2024-06-28
> **Typos**
>
> > Proposition 2: “then, ELBO is invariant”, shouldn’t it be “the gradient of the ELBO is invariant”?
>
> > Proposition 3: There is a “< alpha” missing, both in the equation of the Prop, and in the proof
>
> >“ Beginning of Section 2.2: “negative leaned likelihood”
>
> > Title of Appendix 1.2: “variatonal”
>
> Thank you for these, they have been amended in the revision.

---

> > ### Comment · Reviewer_U1Sv · 2024-07-14
> >
> > Thanks for your edits and your clarifications in the paper!

---

### Decision · Action_Editor_aTUk · 2024-09-21

**Recommendation:** Accept as is

**Comment:**

Despite novelty concerns, I recommend accepting this paper for TMLR publication. The work makes several important contributions to OOD detection for deep generative models: 1. It provides rigorous theoretical justification for using gradient information in OOD detection, demonstrating invariance properties of the score vector. 2. The authors present extensive empirical validations of their proposed approximations across various model types, including diffusion models. IN particular, the evaluation is more comprehensive than prior works, covering multiple training distributions and model types, providing a more robust performance measure. The application to diffusion models extends the method's applicability to state-of-the-art generative models.

While similar methods have been independently discovered, this work offers a unique combination of theoretical insight, empirical validation, and practical applicability. In my opion, the authors have adequately addressed reviewer concerns, clarifyng relationships to classical invariance results and providing more detailed explanations of certain concepts.

I believe this work makes a sufficiently significant contribution to OOD detection for deep generative models and is suitable for TMLR publication.

**Audience:**

The reviewers unanimously agree that this paper would interest the TMLR audience. The work tackles a significant challenge in machine learning (BFSA), providing comprehensive theoretical and empirical analysis on OOD detection using gradient information from deep generative models (SnJ5). The novel applications to various model types, including diffusion models (as noted byU1Sv), enrich our understanding of ideas previously explored by Choi et al. (2021) and Bergamin et al. (2022). The paper's clear writing and extensive experiments (SnJ5) further enhance its likely appeal to the TMLR audience, contributing to the broader discourse on deep generative model reliability and interpretability.

**Claims And Evidence:**

The claims in this submission are well-supported by theoretical arguments and extensive empirical evidence. The authors provide a principled approach to OOD detection using gradient information (BFSA), backed by theoretical demonstrations of invariance properties and empirical validations of their approximations. All reviewers agree on the validity of the claims. The use of the FIM to reweigh gradient test points is well-motivated, both heuristically and through an invariance argument (U1Sv). The paper contributes by evaluating gradient-based methods for OOD detection on deep generative models and connecting this approach to the Fisher Information Metric (SnJ5). While novelty concerns exist due to similar methods being independently discovered (U1Sv, SnJ5), the authors have addressed this by highlighting their unique contributions, including theoretical demonstrations, empirical investigations of approximations, and comprehensive evaluations (SnJ5) across multiple model types, including diffusion models.